## REPORT

# VPS13B is localized at the interface between Golgi cisternae and is a functional partner of FAM177A1

Berrak Ugur[1,2,3,4,5], Florian Schueder[1,6], Jimann Shin[7], Michael G. Hanna[1,2,3,4,5], Yumei Wu[1,2,3,4,5], Marianna Leonzino[1,2,3,5], Maohan Su[1], Anthony R. McAdow[7], Catherine Wilson[8], John Postlethwait[8], Lilianna Solnica-Krezel[7], Joerg Bewersdorf[1,9,10,11], and Pietro De Camilli[1,2,3,4,5]

Mutations in VPS13B, a member of a protein family implicated in bulk lipid transport between adjacent membranes, cause Cohen syndrome. VPS13B is known to be concentrated in the Golgi complex, but its precise location within this organelle and thus the site(s) where it achieves lipid transport remains unclear. Here, we show that VPS13B is localized at the interface between proximal and distal Golgi subcompartments and that Golgi complex reformation after Brefeldin A (BFA)–induced disruption is delayed in *VPS13B* KO cells. This delay is phenocopied by the loss of FAM177A1, a Golgi complex protein of unknown function reported to be a VPS13B interactor and whose mutations also result in a developmental disorder. In zebrafish, the *vps13b* ortholog, not previously annotated in this organism, genetically interacts with *fam177a1*. Collectively, these findings raise the possibility that bulk lipid transport by VPS13B may play a role in the dynamics of Golgi membranes and that VPS13B may be assisted in this function by FAM177A1.

## Introduction

Life in eukaryotic cells relies on the regulated transport of proteins and lipids from one subcellular compartment to another. Lipid transport is achieved both by membrane traffic as well as by a multitude of lipid transport proteins, many of which act at membrane contact sites. Most such proteins function by a piecemeal shuttle mechanism (Saheki and De Camilli, 2017; Balla et al., 2019; Wong et al., 2019; Reinisch and Prinz, 2021). Recently, however, a class of proteins thought to function by a bridge-like mechanism at these sites (hence collectively referred to as bridge-like lipid transfer proteins [BLTPs]) and thus optimally suited for the unidirectional bulk delivery of lipids between two closely apposed membranes has been identified (Levine, 2022; Neuman et al., 2022; Hanna et al., 2023). One such protein is Vps13 (Dziurdzik and Conibear, 2021; Leonzino et al., 2021). While the single yeast Vps13 protein has multiple localizations and functions, the four mammalian VPS13 proteins have distinct, but partially overlapping functions. Mutations in each of them cause severe neurological disorders (Ugur et al., 2020): chorea-acanthocytosis (VPS13A) (Rampoldi et al., 2001), Cohen syndrome (VPS13B) (Kolehmainen et al., 2003), Parkinson's disease (VPS13C) (Lesage et al., 2016; Darvish et al.,

2018; Schormair et al., 2018), and several neurological disorders or embryonic lethality in the case of complete loss-of-function (VPS13D) (Gauthier et al., 2018; Seong et al., 2018; Koh et al., 2020). VPS13A, VPS13C, and VPS13D are localized at contacts between the ER and other organelles (Kumar et al., 2018; Yeshaw et al., 2019; Guillen-Samander et al., 2021; Cai et al., 2022; Hancock-Cerutti et al., 2022), while VPS13B is localized predominantly in the Golgi complex (Seifert et al., 2011), with additional pools on lipid droplets (Du et al., 2023) and possibly at endosomes (Koike and Jahn, 2019), but not, at least so far as currently known, at membrane contacts with the ER. VPS13B is also the most divergent in amino acid sequence among the four paralogs (Levine, 2022).

In agreement with localization of VPS13B in the Golgi complex (Seifert et al., 2011, 2015), a less compact Golgi structure and an altered N-glycosylation pattern were observed in cells from Cohen syndrome patients and in several mammalian *VPS13B* knock-out (KO) and knock-down cell lines (Seifert et al., 2011; Duplomb et al., 2014; Zorn et al., 2022). *Vps13b* KO mice were reported to have moderate neuroanatomical defects (Montillot et al., 2023) and male sterility, as Vps13b is required for the

[1]Department of Cell Biology, Yale University School of Medicine, New Haven, CT, USA; [2]Department of Neuroscience, Yale University School of Medicine, New Haven, CT, USA; [3]Program in Cellular Neuroscience, Neurodegeneration, and Repair, Yale University School of Medicine, New Haven, CT, USA; [4]Aligning Science Across Parkinson's Collaborative Research Network, Chevy Chase, MD, USA; [5]Howard Hughes Medical Institute, Yale University School of Medicine, New Haven, CT, USA; [6]Department of Microbial Pathogenesis, Yale University School of Medicine, New Haven, CT, USA; [7]Department of Developmental Biology, Washington University School of Medicine, St. Louis, MO, USA; [8]Institute of Neuroscience, University of Oregon, Eugene, OR, USA; [9]Nanobiology Institute, Yale University, West Haven, CT, USA; [10]Department of Biomedical Engineering, Yale University, New Haven, CT, USA; [11]Department of Physics, Yale University, New Haven, CT, USA.

Correspondence to Pietro De Camilli: pietro.decamilli@yale.edu

M. Leonzino's current affiliation is the Institute of Neuroscience, Consiglio Nazionale delle Ricerche, Milan, Italy.

growth of the acrosomal membrane of sperm, a Golgi complex–derived membrane critical for fertilization (Da Costa et al., 2019). However, the precise localization of VPS13B within the Golgi complex and thus the sites where its lipid transport function is likely achieved remain unknown.

Here, we provide evidence that VPS13B is localized between proximal and distal Golgi complex compartments. We demonstrate that loss of VPS13B in HeLa cells delays Golgi complex recovery after its Brefeldin A (BFA)–induced dispersion, raising the possibility that VPS13B's lipid transfer function may have a role in Golgi complex reformation. We further provide evidence in mammalian cells and in zebrafish for a functional partnership between VPS13B and FAM177A1, a recently identified Golgi complex protein (Fasimoye et al., 2023; Hickey et al., 2023; Kohler et al., 2024) of unknown function implicated in a human neurodevelopmental disorder (Alazami et al., 2015; Kohler et al., 2024).

## Results and discussion

### Predicted structure of VPS13B

Comparative analysis of the primary sequences of the four mammalian VPS13 proteins reveals that VPS13B is the most divergent among them (Velayos Baeza et al., 1993; Levine, 2022). Such divergence occurred early in evolution, as in protists and algae, whose genome includes only two VPS13 genes, one of these two genes encodes a protein with sequence similarity to VPS13B, while the other appears to be the ancestor of VPS13A, VPS13C, or VPS13D. In spite of the primary sequence divergence, fold prediction algorithms (Yang et al., 2020; Jumper et al., 2021) reveal that VPS13B has all the defining features of VPS13 family proteins: a rod-like core (~27-nm long) comprising 13 repeating β groove (RBG) motifs (Levine, 2022) flanked at its C-terminal portion by folded domains with targeting and likely regulatory functions (Fig. 1 A; and Fig. S1, A and B). One distinct feature of VPS13B is the presence of an accessory folded domain, a module with a jelly-roll fold (Dall'Armellina et al., 2022; Levine, 2022; Hanna et al., 2023), which is an outpocketing of the RBG rod just upstream of the VAB (Vps13 Adaptor Binding) domain (shown in gray in Fig. 1 A and Fig. S1, A–C). VPS13A, VPS13C, and VPS13D lack this domain but have instead a WWE (Trp-Trp-Glu) domain (VPS13A and VPS13C) and a ricin-B domain (VPS13D) (Hanna et al., 2023).

### VPS13B is localized at the interface between proximal and distal Golgi membranes

Due to low levels of endogenous protein expression and inconsistencies with commercially available antibodies for immunocytochemistry, to gain further insight into the subcellular targeting of VPS13B, we expressed codon-optimized versions of full-length human VPS13B, fused to either GFP (VPS13B^GFP) or Halo (VPS13^Halo), in COS7 cells. As we had observed with other VPS13 proteins (Guillen-Samander et al., 2021; Cai et al., 2022), codon optimization helps in achieving robust expression of these very large proteins. GFP or Halo was inserted into the unstructured loop that connects the fifth and sixth repeating RBG motifs, a position predicted not to interfere with the hydrophobic

channel and thus with lipid transport activity (Fig. S1, A and D). In agreement with previous reports (Seifert et al., 2011, 2015), VPS13B^GFP partially colocalized with immunoreactivity for the cis-Golgi protein GM130 (Fig. 1 B). However, the fluorescence patterns of the two proteins did not completely overlap. To gain more precise insight into the localization of VPS13B within the Golgi complex, we used three different strategies.

In one strategy, we used 4Pi-single-molecule switching (SMS) super-resolution fluorescence microscopy, which allows imaging with an isotropic resolution of ~10–20 nm with immunofluorescence labeling specificity (Zhang et al., 2020). In VPS13B^GFP expressing HeLa cells immunolabeled for GFP and for cis- (GM130) and trans- (TGN46) Golgi compartments, we observed that VPS13B^GFP is enriched in the cis-medial Golgi area (Fig. 1 C and Video 1). In addition, we noticed that the VPS13B signal was more punctate than the GM130 or TGN46 signals, in line with what is expected from a membrane contact site protein.

As a second strategy, we performed 7-target super-resolution FLASH-PAINT (fluorogenic labeling in conjunction with transient adapter-mediated switching for high-throughput DNA-PAINT) imaging (Schueder et al., 2024), a variant of DNA-PAINT (DNA points accumulation for imaging in nanoscale topography) (Jungmann et al., 2010; Schnitzbauer et al., 2017), enabling efficient multiplexed imaging. FLASH-PAINT utilizes transient binding of short diffusing fluorescently labeled DNA oligomers (termed "imagers") to single-stranded DNA docking sites at the target (in our case antibodies to specific proteins) via a transient adapter, resulting in apparent fluorescent blinking at these sites (Fig. 1 D). In a subsequent analysis, super-resolved coordinates of the docking sites are extracted from the observed distinct single-molecule blinking events.

To enable sequential multiplexed imaging of VPS13B^GFP (using anti-GFP nanobodies) relative to different Golgi complex proteins, seven protein species were each labeled with an antibody/nanobody conjugated to a different single-stranded DNA docking site. For imaging, a transient adapter was used to mediate the interaction between the imager and the docking site. To switch from imaging one target species to the next, a single-stranded DNA eraser (Fig. 1 D) was added after the completion of each round of imaging. This eraser binds to the adapter from the previous round, preventing the adapter from binding to the docking site and, therefore, effectively eliminating the signal. This procedure was then repeated for all target antigens. An analysis of distances of the various antigens from each other revealed that VPS13B was closest to the cis-Golgi complex markers GM130 (on average ~28 nm) (Fig. 1, E–E″) and GRASP65 (on average ~33 nm) (Fig. 1, G–G″), and the furthest away from trans-Golgi markers TGN46 and Golgin97 (Fig. 1, E–F″ and H; and Tables S7 and S8). VPS13B was also at some distance from β'-COP (coat protein) positive signal (Fig. 1, F–F″ and H).

As a third strategy, we exposed live cells to a hypotonic shock. This treatment rapidly expands the volume of the cytoplasm, thus decreasing organelle crowding and causing partial release of cytosolic proteins. It also causes membrane organelles

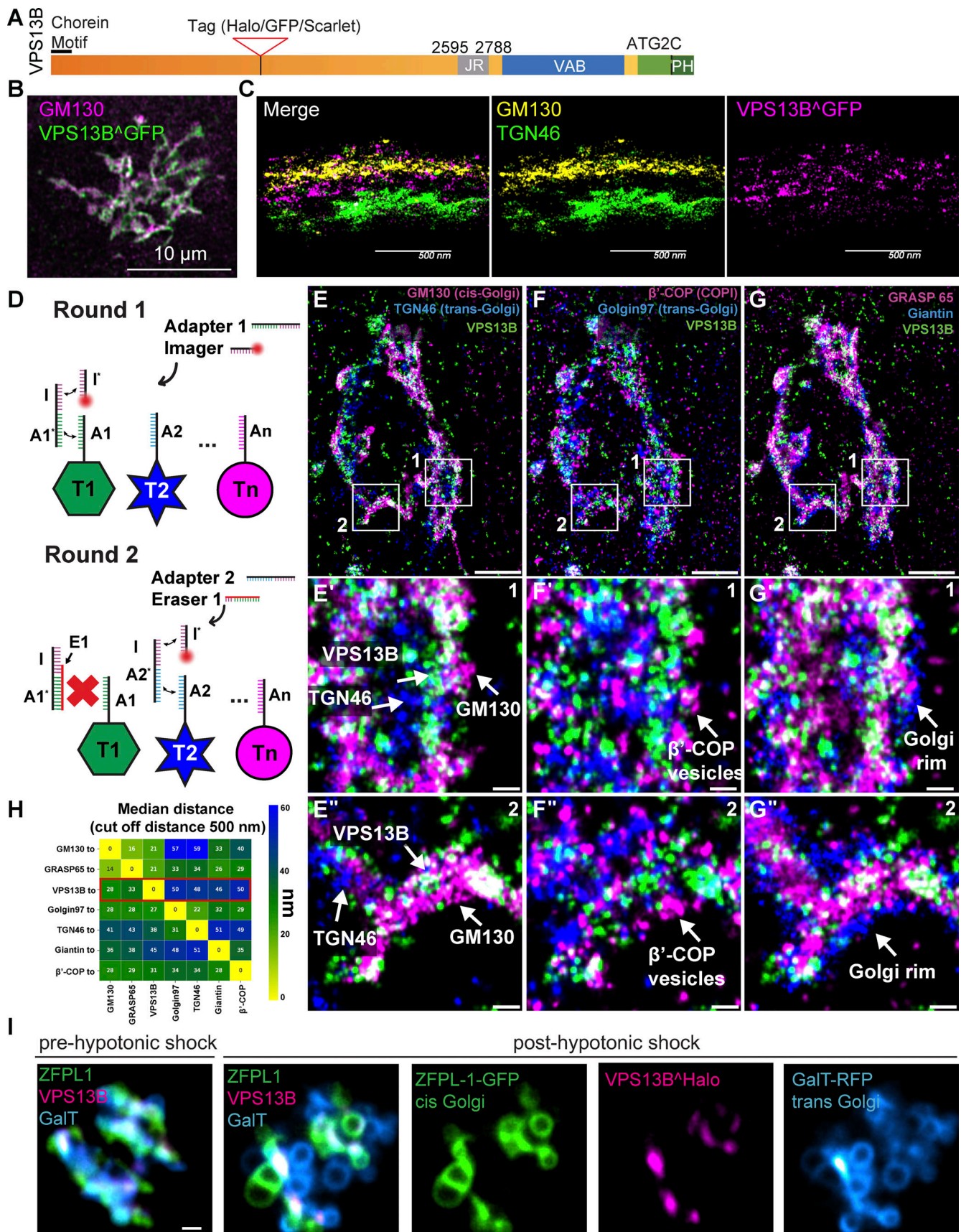

Figure 1. **VPS13B is localized at the interface between cis-Golgi and trans-Golgi membranes. (A)** Domain cartoon of human VPS13B. **(B)** COS7 cells expressing codon-optimized human VPS13B^GFP immunolabeled for GFP and GM130. Scale bar = 10 μm. **(C)** Cross-section (side view) of HeLa cells expressing

codon-optimized human VPS13B^GFP immunolabeled for GFP, GM130, and TGN46 imaged using 4Pi SMS microscopy. Scale bar = 500 nm. **(D)** Schematic representation of labeling steps used for FLASH-PAINT. FLASH-PAINT performed in HeLa cells expressing VPS13B^GFP and immunolabeled with anti-GFP. **(E–G)** FLASH-PAINT signals of a Golgi complex immunolabeled for the indicated Golgi complex targets. Scale bar = 2 μm. High magnification fields of the boxed areas in fields E–G are shown below in fields E'–G' and E''–G''. Scale bar = 250 nm. **(H)** Median distances between super-resolved signals of different targets. Only signals closer than 500 nm to each other were considered. **(I)** Snapshots of COS7 cell expressing VPS13B^halo, ZFPL1-GFP (a cis-Golgi marker), and GalT-RFP (a trans-Golgi marker) before and after (8 min) hypotonic shock. Scale bar = 1 μm.

to swell, but, at least within short time ranges (minutes), does not disrupt their membrane contacts, which in fact may coalesce into fewer and larger contacts (King et al., 2020; Guillen-Samander et al., 2021). As a result, the localization of proteins at membrane contact sites can be more easily visualized than in intact cells. We carried out these experiments in COS7 and HeLa cells expressing VPS13^Halo and, in pairwise combinations: ZFPL1-GFP (a cis-Golgi marker [Chiu et al., 2008]), GalT-RFP (a trans-Golgi marker [Roth and Berger, 1982]), ManII-RFP (a medial Golgi marker [Velasco et al., 1993]), and Bet1-GFP (a cis-Golgi and ER-Golgi intermediate compartment associated protein [Hay et al., 1998]). We found that upon exposure of cells to strong hypotonic conditions, swollen Golgi cisternae coalesce primarily in two distinct sets of closely apposed vacuoles in which cis- and trans-Golgi markers are differentially segregated but which are both positive to variable extent for the medial Golgi marker (Fig. 1 I, Video 2, and Fig. S1 E). Interestingly, the bulk of VPS13B localized at the interface of the two sets of vacuoles, suggesting that it may function as one of the bridges tethering them (Fig. 1, C, E', and I; and Fig. S1 E). We also performed similar experiments with cells expressing tagged VPS13B and markers of the ER and of the Golgi complex but did not find convincing evidence for the presence of VPS13B at the interface of ER and Golgi membranes (Fig. S1 F), where other lipid transport proteins are known to function (De Matteis et al., 2007; Venditti et al., 2020). This is consistent with the lack of an FFAT (two phenylalanines (FF) in an acidic tract) motif for ER binding in VPS13B, a motif that is present in VPS13A, VPS13C, and VPS13D (in the latter case a phosphoFFAT motif [Guillen-Samander et al., 2021]). As a low probability phospho-FFAT motif is predicted in the N-terminal region of VPS13B (Di Mattia et al., 2020), we tested the N-terminal fragment of VPS13B that comprises this predicted motif (a.a. 1–586) fused to GFP and observed a cytosolic distribution (Fig. S1 G). A phosphomimetic VPS13B (VPS13B S559E, S560E, S563E) fragment also displayed a cytosolic distribution when co-expressed with VAPB-RFP (VAMP associated protein B) (Fig. S1 G), thus ruling out that VAPB may be present in limiting concentration and indicating that N-terminal fragment of VPS13B is not recruited to the ER unlike other VPS13 proteins.

While performing these localization studies, we noticed that overexpressed VPS13B can also localize to lipid droplets, more so when the temperature is lowered to 20°C. A previous study indicated that VPS13B can be present at the Golgi complex–lipid droplet interface (Du et al., 2023), although our results indicate that VPS13B can localize to lipid droplets independent of their proximity to the Golgi complex (Fig. S1 H). Of note, Du et al. also report no evidence for a recruitment of VPS13B to the ER via VAPB.

## FAM177A1 is a neighbor of VPS13B in the Golgi complex

Previously, the localization of VPS13B at the Golgi complex was reported to be dependent on Rab6 (Seifert et al., 2015), a Rab family member with multiple reported sites of action in the Golgi complex (Feldmann et al., 1995). Further elucidation of the properties and physiological function of VPS13B will be helped by the identification of potential functional partners. These proteins may assist VPS13B in its tethering function or lipid extraction and delivery from and to membranes. Recently, we have shown that FAM177A1, a protein of unknown function identified as a potential interactor of VPS13B by several high-throughput proteomic studies (Huttlin et al., 2015, 2017, 2021) (Fig. S2 A) is localized in the Golgi complex (Fasimoye et al., 2023; Kohler et al., 2024) (Fig. S2 B). Interestingly, recessive loss-of-function mutations in *FAM177A1* are responsible for a neurodevelopmental disorder that displays characteristics similar to those of Cohen syndrome, as reported in Alazami et al. (2015) and supported by additional cases (Kohler et al., 2024). This raises the possibility that FAM177A1 may be a functional partner of VPS13B. FAM177A1 belongs to a highly conserved FAM177 eukaryotic protein family (Interpro: IPR028260), and humans express two FAM177A1 isoforms (referred to as isoform 1 and 2), which result from two alternative initiation sites and thus differ in their N-terminal region because of the exclusion or inclusion of a 23-a.a. fragment. Based on folding predictions (Jumper et al., 2021; Varadi et al., 2022), both FAM177A1 isoforms are primarily disordered proteins with two highly conserved α-helices and a short β-sheet hairpin (Fig. 2 A), without transmembrane regions.

Both isoforms, when tagged with GFP (we did not detect any specific localization with the commercially available antibodies), colocalize with tagged-VPS13B in the Golgi complex (Fig. 2 B), although at high level of expression, some signals can also be observed throughout the ER, where VPS13B is never observed. In further studies, we used primarily isoform 1. We found that its localization in the Golgi complex does not depend on VPS13B because its localization is preserved in VPS13B KO cells (Fig. S3 B). Fractionation of the cell lysate into a cytosolic and a total membrane fraction, followed by western blotting for GFP, revealed that the bulk of FAM177A1-GFP is membrane-associated, but that a small amount remained soluble, while western blotting of the same fractions for a control cytosolic protein (GAPDH) and a control intrinsic membrane protein (STX6) showed that they exclusively segregated as expected in the soluble and membrane fractions, respectively (Fig. 2 C). These findings confirmed that FAM177A1 is not an intrinsic membrane protein, although it interacts with membranes.

To determine how FAM177A1 is targeted to the Golgi complex, we transfected COS7 cells with constructs encoding the

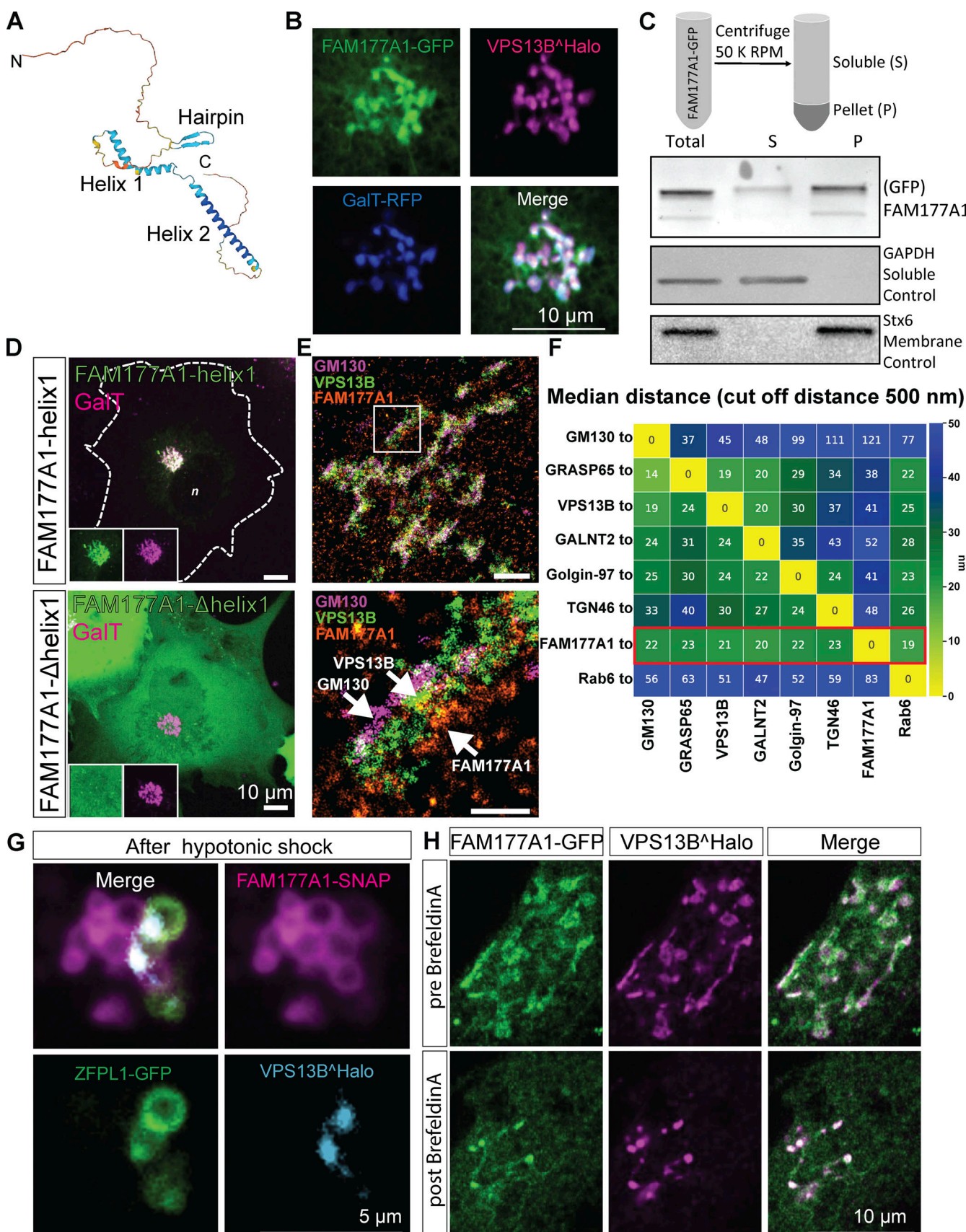

Figure 2. **FAM177A1 is a protein neighbor of VPS13B in the Golgi complex. (A)** Alphafold2 prediction of human FAM177A1 structure. **(B)** COS7 cells expressing VPS13B^Halo, FAM177A1-GFP, and GalT-RFP showing that FAM177A1 is localized at the Golgi complex. **(C)** Western blot analysis of soluble cytosolic

fraction and total membrane faction of HeLa cells expressing FAM177A1-GFP. GAPDH was used as a soluble control, Stx6 as a membrane control protein. **(D)** Top panel: Co-expression of FAM177A1-helix1-GFP and GalT-RFP in COS7 cells. Bottom panel: Co-expression of FAM177A1-Δhelix 1-GFP and GalT-RFP in COS7 cells. Scale bar = 10 µm. **(E)** FLASH-PAINT performed in HeLa cells expressing VPS13B^GFP and FAM177A1-Halo and immunolabeled with anti-GFP, anti-halo, and anti-GM130 antibodies. **(F)** Median distances between super-resolved signals of different Golgi complex targets. Only signals closer than 500 nm to each other were considered. **(G)** Snapshots of COS7 cells expressing FAM177A1-SNAP, ZFPL1-GFP, and GalT-RFP treated with water for 10 min. Scale bar = 5 µm. **(H)** Snapshots of HeLa cells expressing FAM177A1 GFP and VPS13B^Halo before and after BFA (5 µg/ml for 50 min) treatment. Scale bar = 10 µm. Source data are available for this figure: SourceData F2.

three predicted folded domains of FAM177A1 fused to GFP and assessed their localizations. Helix 1 (a.a. 118–147 in isoform 1 and a.a. 95–124 in isoform 2) predominantly localized to the Golgi complex (Fig. 2 D, top panel), whereas Helix 2 (a.a. 156–197 in isoform 1 and a.a. 133–174 in isoform 2) and the β-sheet hairpin (a.a. 78–90 in isoform 1 and 55–67 in isoform 2) were cytosolic (Fig. S2 C). Moreover, a FAM177A1 construct lacking Helix 1 also had a diffuse cytosolic localization (Fig. 2 D), providing further evidence that Helix 1 is responsible for the Golgi complex localization of FAM177A1. FAM177A1 has a close paralog in the human genome, FAM177B (Fig. S2 D). Expression of FAM177B-flag in HeLa cells showed that this protein also localizes to the Golgi complex, suggesting that the two proteins have overlapping functions (Fig. S2 E). Consultation of the Protein Atlas and RNA sequencing (RNA-seq) depositories (https://www.proteinatlas.org/ENSG00000197520-FAM177B) reveals that FAM177A1 has a broad distribution across tissues, while FAM177B has a more restricted expression and is generally expressed at much lower levels, with highest expression reported in the stomach.

To obtain further insight into the localization of FAM177A1, we carried out FLASH-PAINT imaging of FAM177A1 relative to seven Golgi complex markers and found that FAM177A1 has a broad distribution throughout the Golgi complex (Fig. 2, E and F; and Fig. S2 F). We also examined the redistribution of FAM177A1-SNAP in response to hypotonic shock in cells also expressing VPS13B^Halo and the cis-Golgi marker ZFPL1-GFP. After the hypotonic shock, FAM177A1-SNAP was present on both cis- and trans-Golgi complex positive vacuoles (Fig. 2 G and Fig. S2 G), whereas ZFPL1-GFP remained confined to one set of them, and VPS13B, as shown in Fig. 1 I, was restricted to the interface between two sets of vacuoles (Fig. 2 G). As the localization of VPS13B was shown to be disrupted by the expression of dominant-negative Rab6a (Rab6a$^{T27N}$) (Seifert et al., 2015), we also investigated whether this Rab6 construct affected the localization of FAM177A1. We found, however, that FAM177A1 remained in the Golgi complex upon Rab6a$^{T27N}$ expression (Fig. S2 H).

We additionally performed experiments with BFA, a fungal metabolite that blocks transport between the ER and Golgi complex by inhibiting Arf1 and induces the reversible disassembly of the Golgi complex (Lippincott-Schwartz et al., 1989). In agreement with its Golgi complex localization, VPS13B was shown to disperse by cell treatment with BFA (Duplomb et al., 2014; Seifert et al., 2015) although we found that few spots remained in the region previously occupied by the Golgi complex. When cells co-expressing FAM177A1 and VPS13B were exposed to BFA, both proteins acquired a diffuse distribution, but the remaining VPS13B puncta were now also positive for FAM177A1

(Fig. 2 H and Video 3). Moreover, when cells expressing VPS13B and GalT were treated with BFA, GalT dispersed into the ER as anticipated, but some GalT positive spots overlapping with the remaining VPS13B puncta were also observed (Fig. S2 I).

## Loss of VPS13B and FAM177A1 leads to delay in Golgi complex reformation after BFA treatment

It was described previously that the loss of VPS13B results in a less compact Golgi structure (Seifert et al., 2011). We generated *VPS13B* KO HeLa cells by CRISPR/Cas9 (Fig. S3 A; and Fig. 3, A and B) and made similar observations using GM130 as a marker (Fig. 3 A). FAM177A1 was still localized in the Golgi in these cells (Fig. S3 B), speaking against the role of VPS13B in its recruitment. Moreover, when *VPS13B* KO cells were exposed to a hypotonic shock we observed that distinct vacuoles where the bulk of cis- and trans-Golgi markers segregated were still in close apposition, as observed in WT cells, ruling out a requirement of VPS13B for their tethering (Fig. S3 E). Given the putative function of VPS13B in the bulk transport of phospholipids between membranes, we explored whether its absence leads to perturbation of Golgi complex dynamics, as assessed by GM130 immunofluorescence. Treatment of VPS13B KO cells with BFA-induced dispersion of the Golgi complex with a time course similar to control cells (Fig. 3 A and Fig. S3 F). However, Golgi complex reformation was delayed in KO cells relative to control cells upon BFA washout. While the Golgi complex fully reassembled within 2 h in control cells, its reassembly was still incomplete in about 50% of the KO cells after 5 h (Fig. 3, A and D). Similar results were observed when the localization of Golgi markers different from GM130 (ZFPL1 and Man II) were monitored (Fig. S3, G and H). This delay was rescued by the exogenous expression of VPS13B^GFP. To determine whether the lipid transport function of VPS13B is important for this rescue, we generated a putative lipid transport dead mutant of human VPS13B (L65K, I81E, L90E, I155R, L169E, A176E, I203R, L238D, I355K, L264R named as LTD Mut1) based on the one generated in *S. cerevisiae* (Li et al., 2020) by overlaying AlphaFold2 predicted structures. Unfortunately, this construct formed aggregates in cells (see Fig. S3 I) and thus could not provide useful information.

We also examined whether loss of *FAM177A1* had an impact on Golgi complex structure and to this aim, we generated *FAM177A1* KO HeLa cells (Fig. 3, A and C; and Fig. S3 C). In these cells, the localization of VPS13B was not affected (Fig. S3 D) and overall Golgi complex morphology was not severely perturbed with the exception of a less compact structure as assessed by GM130 staining (Fig. 3 A). Moreover, dispersion of the Golgi complex upon BFA treatment was not different from controls. However, as in *VPS13B* KO cells, Golgi complex recovery was delayed, and

Figure 3. **Loss of VPS13B and FAM177A1 leads to delay in Golgi complex reformation after BFA treatment. (A)** Anti-GM130 immunofluorescence of WT, *VPS13BKO[1]*, *VPS13BKO[2]*, *FAM177A1KO*, and *FAM177A1; VPS13B* DKO cells before incubation with BFA, after 1 h in BFA (5 µg/ml), and after subsequent washings as

indicated; superscripts indicate different clones. Scale bar = 10 μm. **(B and C)** Western blots of total cell homogenates showing loss of the VPS13B band and/or the FAM177A1 band in the KO clones. GAPDH was used as a gel loading control. **(D)** Quantification of Golgi complex reformation in cells with the indicated genotypes after BFA washout for 2 or 5 h. "Rescue" refers to the exogenous expression of the knocked-out protein. Data are mean ± SEM $n$ = 3 per condition, in each condition 75–100 cells were quantified. Unpaired, two-tailed $t$ tests. NS, not significant. ****$P < 0.0001$; ***$P < 0.001$; **$P < 0.01$; *$P < 0.05$. Source data are available for this figure: SourceData F3.

this delay was rescued by exogenous expression of *FAM177A1* (Fig. 3, A and D).

Finally, we generated *VPS13B* and *FAM177A1* double-KO HeLa cells (DKO, Fig. 3 A and C). In these cells, Golgi complex recovery following BFA was even more strongly delayed when compared to the recovery in both single KO cells. Following BFA washout, ~50% of *FAM177A1* KO cells reformed their Golgi complex by 5 h, as shown before for *VPS13B* KO cells (Fig. 3 D; and Fig. S3, G and H). However, only <30% of the *VPS13B;FAM177A1* DKO cells displayed recovery of the Golgi complex 5 h after BFA washout, suggesting a synergistic effect of the two mutations (Fig. 3, A and D; and Fig. S3, G and H).

### A partnership of fam177a1 and vps13b in zebrafish

To test whether our results in cultured cells had relevance to organism physiology, we carried out studies in zebrafish. Zebrafish has two FAM177 genes, *fam177a1a* and *fam177a1b*. In contrast, although *Cyprinus carpio* encodes a *Vps13b*, a gene encoding VPS13B ortholog had not been annotated in zebrafish. As a functional partnership between Fam177a1 and Vps13b implies their coexpression, we searched for a *VPS13B* ortholog in the unannotated zebrafish genome sequence by BLAST searches (http://blast.ncbi.nlm.nih.gov/Blast.cgi, RRID:SCR_004870) and identified a *vps13b* candidate ortholog. Analysis of its structure revealed strong similarity to mammalian VPS13B, including the presence of a jelly-roll module (Fig. 3 A). Furthermore, the candidate *vps13b* gene showed conserved synteny with the human VPS13B gene (Fig. S3 J). The conservation of sequence and genomic location shows that these two genes are indeed orthologs. In addition, we performed quantitative RT-PCR (qRT-PCR) analysis of *vps13b* transcripts in WT zebrafish embryos at early developmental stages (two-cell, 6 and 24 h post-fertilization [hpf]) and confirmed the expression of *vps13b* at all these stages, with higher levels at the two-cell stage indicating maternal expression (Fig. S3 K).

Recently, we generated large deletions in the two zebrafish *FAM177A1* homologs, *fam177a1a* and *fam177a1b* (*fam177a1a/b*) (Kohler et al., 2024). While fish with these mutations, *fam177a1a; fam177a1b* DKO were viable and fertile, they exhibited a delay in reaching WT length during embryogenesis and at 25 hpf were significantly shorter than controls (Fig. 4, B and C, left panels) (Kohler et al., 2024). To determine if *fam177a1a/b* functionally interacts with zebrafish *vps13b*, we injected WT and *fam177a1a; fam177a1b* DKO embryos at the one-cell stage with two *vps13b* guide RNAs and Cas9 to generate a pool of edited fish ("crispants" [Hwang et al., 2013]) (Fig. S3 L). As these injections can only impair zygotic gene function and in a mosaic fashion, the resulting phenotypes would be expected to be a hypomorph phenotype rather than a complete loss-of-function phenotype. Crispant fish embryos for *vps13b* displayed regular body length at

all stages (Fig. 4, B and C, left panels), whereas *fam177a1a;fam177a1b* DKO;*vps13b* crispants had significantly shorter (25% reduction compared to WT) body length at 25 hpf when compared with *fam177a1a;fam177a1b* DKO fish at the same stage (15% reduction compared to WT) indicating a functional partnership between Fam177a1 proteins and Vps13b (Fig. 4, B and C, left panels). Moreover (compare right and left panels in Fig. 4 C), when zebrafish embryos were continuously exposed to BFA from 6 to 25 hpf (0.8 μg/ml, a dose comparable to the concentration used for in vitro studies), a treatment that by itself results in a 7% reduction in body length of WT animals at 25 hpf, an additional decrease in body length was observed in *fam177a1a; fam177a1b* DKO (30% reduction compared with untreated WT) and *vps13b* crispants (20% reduction compared with untreated WT) revealing that loss of these proteins sensitizes zebrafish embryos to BFA effects. Although the *fam177a1a;fam177a1b* DKO; *vps13b* crispants treated with BFA had a significantly decreased body length (35% reduction) when compared with either WT animals or *vps13b* crispants alone, they did not show a significant decrease when compared with *fam177a1a;fam177a1b* DKO animals. In sum, loss of either *vps13b* or *fam177a1a/b* in zebrafish leads to a BFA sensitivity in animals and the hypomorphic zygotic *vps13b* crispant loss of function enhances *fam177a1a/b* mutant phenotype, suggesting a functional interaction of these two Golgi-localized proteins.

### Concluding remarks

Our study provides new evidence for the idea that VPS13B differs in some special way from the other three mammalian VPS13 proteins, in agreement with its early evolutionary divergence from the common progenitor of VPS13A, VPS13C, and VPS13D (Velayos-Baeza et al., 2004; Levine, 2022). In contrast to what we have found for the other three VPS13 proteins, we did not find evidence for the binding of VPS13B to the ER or, more specifically, at the interface of the ER with other organelles. This negative result is supported by the lack of an FFAT motif in its sequence. The absence of an FFAT motif and lack of evidence for ER binding was also observed for another member of the RBG lipid transport protein family, SHIP164/BLTP3B (Hanna et al., 2022). Of note, the N-terminal region of VPS13B, the so-called chorein motif, which is the most conserved region among VPS13 family members, has a different overall charge than the chorein motifs of VPS13A, VPS13C, and VPS13D (Levine, 2022). Using 4Pi-SMS microscopy (Zhang et al., 2020) and FLASH-PAINT, a recently developed super-resolution microscopy technique (Schueder et al., 2024), we have localized the protein in the cis-region of the Golgi complex, as previously reported (Seifert et al., 2011) but also extending toward the medial region. Furthermore, using hypotonic swelling as an experimental system to visualize membrane contact sites within the Golgi complex,

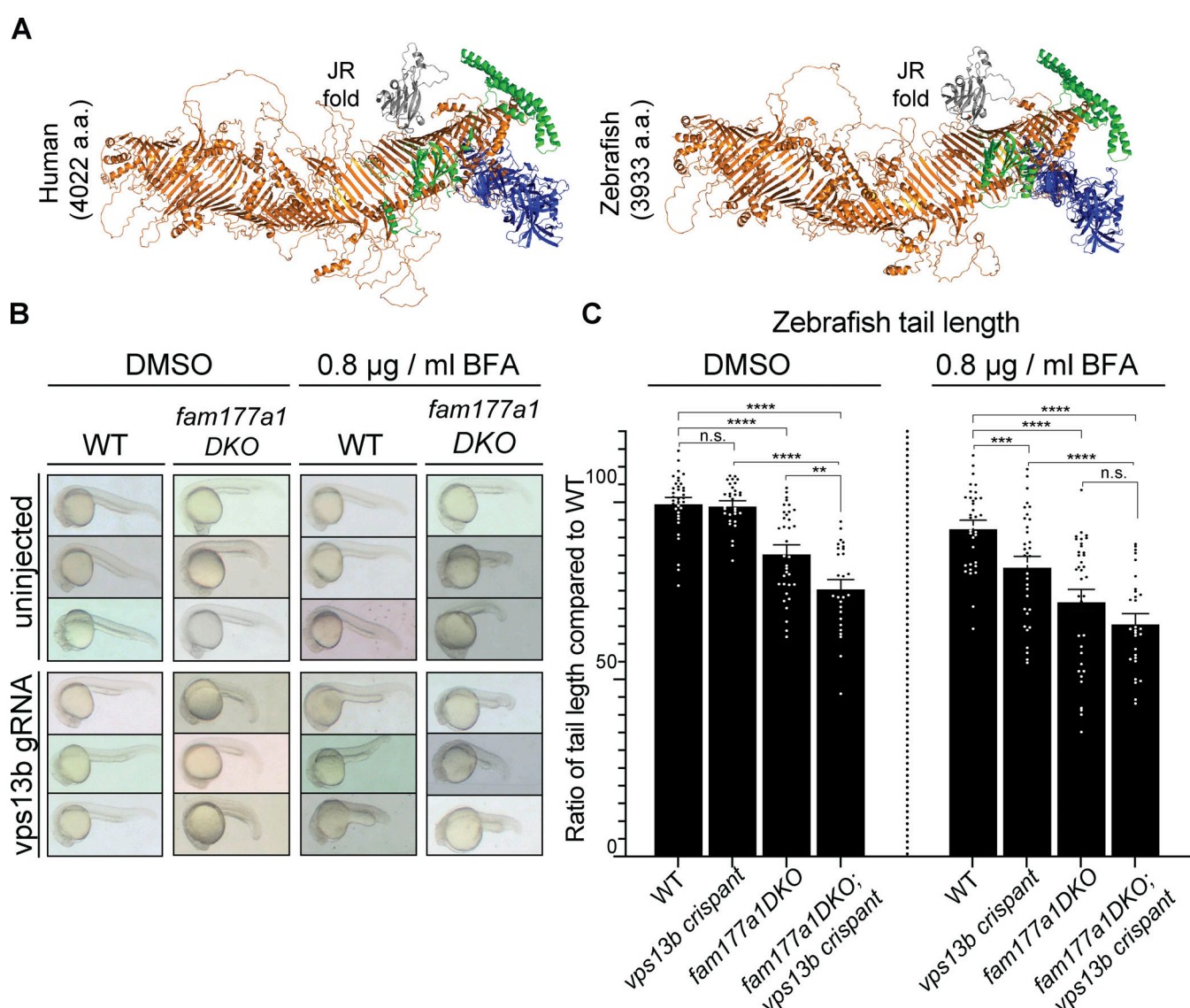

Figure 4. **A partnership of *fam177a1* and *vps13b* in zebrafish. (A)** Alphafold2 prediction of human VPS13B and zebrafish Vps13b. **(B)** Representative images of WT and *fam177a1a;fam177a1b DKO (fam177a1DKO)* zebrafish embryos that are either uninjected or injected with *vps13b* gRNAs/Cas9. The left panel shows embryos treated with DMSO at 6 hpf until 25 hpf whereas right panel shows embryos treated with 0.8 μg/ml BFA at 6 hpf until 25 hpf. **(C)** Quantifications of tail length results shown in B. Data are shown as means ± SEM, Unpaired, two-tailed *t* tests. n.s., not significant. ****P < 0.0001; ***P < 0.001; **P < 0.01.

we obtained evidence for the presence of VPS13B at contacts between different Golgi membranes. Interestingly, we found that such treatments result in the coalescence of Golgi cisternae primarily in two distinct sets of vacuoles where cis- (Bet1 and ZFPL) and trans- (GalT) markers differentially segregated while partially intermixing with a medial Golgi marker (Man II). In such vacuole clusters, VPS13B accumulated selectively at contacts between the two sets of vacuoles, consistent with a tethering and bridge-like lipid transport function of this protein. Our study also shows that the lack of VPS13B results in a delay in the re-formation of the Golgi complex after its BFA-induced dispersion. While this finding may have many explanations, a defect in Golgi cisternae reassembly due to a defect in lipid transport between Golgi complex subcompartments is plausible.

A membrane tethering function for VPS13B implies a minimum of two binding sites, one on each of the two membranes connected by this protein. At least one binding site is likely to be Rab6, as VPS13B has been reported to be a Rab6 effector, and dominant-negative Rab6 was shown to result in a cytosolic localization of VPS13B (Seifert et al., 2015). This is surprising because, upon loss of Rab6 binding, VPS13B could be expected to remain associated with the other membranes via its second binding site. It is possible that loss of Rab6 binding may allow the occlusion of the second binding site through an intramolecular interaction. A similar result was obtained for VPS13C, which has binding sites for Rab7 on lysosomes and for VAP on the ER, but becomes diffusely cytosolic upon expression of dominant negative Rab7 (Hancock-Cerutti et al., 2022).

Our study also suggests a functional link of VPS13B to FAM177A1, a hit in proteome-wide screens of VPS13B interactors (Huttlin et al., 2015, 2017, 2021). Our current study provides strong evidence for a close proximity of pools of the two proteins in the Golgi complex, although, so far, we do not have evidence for a direct interaction between them. FAM177A1 is not needed for the Golgi complex localization of VPS13B and, vice versa, FAM177A1 localizes to the Golgi complex even in the absence of VPS13B. Additionally, as shown by the hypotonic lysis experiments, the localization of FAM177A1 in the Golgi complex is more widespread on Golgi membranes than the localization of VPS13B, suggesting that only a small pool of the two proteins may interact with each other at any given time. On the other hand, we have found that the delay in Golgi complex re-formation after BFA-induced dispersion observed in *VPS13B* KO cells is phenocopied in *FAM177A1* KO cells and that the loss of both proteins results in a more severe delay, consistent with a synergistic effect of the two proteins.

Likewise, an additive effect of VPS13B and FAM177A1 is suggested by experiments in zebrafish, where a *vps13b* gene had not been annotated before. In this organism, the developmental defect caused by the absence of *fam177a1a/b* is enhanced by the partial loss of function of *vps13b* (*fam177a1a fam177a1b DKO;vps13b* crispant fish). Moreover, chronic exposure of zebrafish to BFA enhances the developmental phenotypes of *fam177a1DKO*, both in the presence or absence of the additional defect in VPS13B, and unmasks a developmental defect in *vps13b* crispant animals. Most interestingly, loss of either protein in humans results in developmental disorders that include neurological defects. In line with a partnership of VPS13B and FAM177A1, the *C. elegans* genome, which lacks a *VPS13B* gene, also lacks a gene encoding FAM177A1.

In conclusion, our studies provide new insight into the site of action of VPS13B and identify FAM177A1 as one of its potential functional partners. Given the putative bridge-like lipid transport function of VPS13 family proteins, they suggest that VPS13B may help translocate lipids between Golgi cisternae by vesicular traffic–independent mechanisms. They provide a foundation for further investigations of the cellular (Seifert et al., 2011; Duplomb et al., 2014; Zorn et al., 2022) and organismal (Da Costa et al., 2019; Kim et al., 2019; Gabrielle et al., 2021; Montillot et al., 2023) phenotypes produced by VPS13B loss of function and thus of molecular mechanisms underlying Cohen syndrome.

While this manuscript was in review, a manuscript was posted in BioRxiv reporting an interaction of VPS13B with Sec23IP (Du et al., 2024, *Preprint*). The two manuscripts are in agreement in reporting the localization of VPS13B in the cis-medial Golgi complex and the lack of association of VPS13B with the ER, in contrast to what has been shown for other VPS13 family members. How the interaction of VPS13B with Sec23IP impacts the findings reported here will require additional studies.

## Materials and methods
### Antibodies and reagents
Primary antibodies used were as follows: rabbit VPS13B (24505-1-AP; RRID:AB_2879579; IB (immuno-blot) 1:250; Proteintech);

rabbit FAM177A1 (A303-366A; RRID:AB_10952864; IB 1:1,000; Bethyl Laboraties); mouse GM130 (610822; RRID:AB_398141; IF (immuno-fluorescence) 1:500; BD Bioscience); rabbit GOLGA2/GM130 (ab30637; RRID:AB_732675 IF 1:300; Abcam and 11308-1-AP; RRID:AB_2115327; Proteintech); rabbit TGN46 (ab50595; RRID:AB_2203289; IF 1:200; Abcam and 10598-1-AP; RRID:AB_2113473; Proteintech); mouse GAPDH (40-1246; IB 1:10,000; Proteus); mouse α-Tubulin (T5168; RRID:AB_477579; IB 1:10,000; Sigma-Aldrich); rabbit GFP (A-11122, RRID: AB_221569, IF 1:250; IB 1:1,000; Invitrogen); rabbit Stx6 (110 062; RRID: AB_887854; IB 1:1,000; Synaptic systems); GRASP 65 (ab174834; Abcam); Giantin (HPA011555; RRID:AB_1079010; Sigma-Aldrich); Golgin 97 (HPA044329; RRID:AB_2678897; Atlas antibodies); sheep anti-GALNT2 (AF7507; Novus); COPI (CMIA10) were customary made in the Rothman lab. Secondary antibodies used were goat anti-mouse IgG (926-32210; RRID: AB_621842; LI-COR Biosciences) and goat anti-rabbit IgG (926-68021; RRID: AB_10706309; LI-COR Biosciences).

Cy3B-modified DNA oligonucleotides were custom-ordered from IDT. DNA-labeled nanobodies were obtained from Massive Photonics. Halo and SNAP-tag ligands (JF549, JF646) were kind gifts from L. Lavis (Janelia Research Campus, Ashburn, VA, USA).

Sodium chloride 5 M (cat: AM9759) was obtained from Ambion. Ultrapure water (cat: 10977-015) was purchased from Invitrogen. μ-Slide 8-well chambers (cat: 80807) were purchased from ibidi. Methanol (cat: 9070-05) was purchased from J.T. Baker. Glycerol (cat: 65516-500 ml), protocatechuate 3,4-dioxygenase pseudomonas (PCD) (cat: P8279), 3,4-dihydroxybenzoic acid (PCA) (cat: 37580-25G-F), and (±)-6-hydroxy-2,5,7,8-tetra-methylchromane-2-carboxylic acid (Trolox) (cat: 238813-5 G) were ordered from Sigma-Aldrich. 1× Phosphate Buffered Saline (PBS) pH 7.2 (cat: 10010-023) was purchased from Gibco. Bovine serum albumin (cat: 001-000-162) was ordered from Jackson ImmunoResearch. Triton X-100 (cat: T8787-50ML, RRID:SCR_008988) was purchased from Sigma-Aldrich.

### DNA plasmids
A plasmid containing codon-optimized cDNA encoding human VPS13B, also including mScarlet fluorescent protein after amino acid residue 1301 flanked by BamHI restriction enzyme sites, was generated by and purchased from GenScript Biotech. This plasmid was linearized with BamHI and used to clone VPS13B^EGFP (RRID: Addgene_ 224589) and VPS13B^Halo (RRID: Addgene_ 224584) by In-Fusion Cloning (Takara Bio).

FAM177A1isoform2-GFP (RRID: Addgene_ 224589), FAM177A1-RFP (RRID: Addgene__227677), FAM177A1-Halo (RRID: Addgene_ 224591), and FAM177A1-SNAP (RRID: Addgene_ 224592) were all cloned from FAM177A1 pcDNA3.1+/C-(K)-DYK (GenScript Clone ID:OHu30351D) using the primers in Table S1 by HiFi (NEB) cloning. G-blocks for VPS13B-LTDmut1, FAM177A1-helix1, FAM177A1-helix2, and FAM177A1-hairpin were cloned using HiFi cloning (described in Table S1). VPS13B N-terminal fragments were generated using primers listed in Table S1 and were cloned into pEGFPN1 (GenBank Accession #U55762) using SacI, AgeI restriction enzyme sites. ZFPL1-GFP (RRID: Addgene_ 224590) and BET1_OHu30430C_pcDNA3.1(+)-C-eGFP

(RRID: Addgene_227676) were purchased from GenScript (Clone ID: OHu29409C, Accession No.: NM_006782.3, and Clone ID:OHu30430C, respectively). GalT-RFP was previously cloned from Addgene plasmid # 11929; RRID:Addgene_11929 and VapB-RFP construct was explained in Dong et al. (2016). ManII-RFP was a kind gift from Rothman Lab (Yale University, New Haven, CT, USA).

## Cell culture and transfection
Cells were cultured at 37°C and 5% $CO_2$ in Dulbecco's Modified Eagle Medium containing 10% fetal bovine serum, 100 U/ml penicillin, 100 mg/ml streptomycin, 2 mM L-glutamine (all from Gibco), and plasmocin prophylactic 5 μg/ml (Invivogen). COS7 (RRID:CVCL_0224) and HeLaM (RRID:CVCL_R965) cells for imaging experiments were seeded on glass-bottomed dishes (MatTek) at a concentration of 50–75 × 10³ cells per dish, transiently transfected using FuGene HD (Promega), and imaged ~48 h later.

A detailed description of cell culture, transfection, immunocytochemistry, and imaging can be found in protocols.io DOI: https://doi.org/10.17504/protocols.io.eq2lyp55mlx9/v1.

## Microscopy
### Live cell imaging
Just before imaging, the growth medium was removed and replaced with prewarmed Live Cell Imaging solution (Life Technologies). Imaging was carried out at 37°C and 5% $CO_2$. Spinning-disk confocal microscopy was performed using an Andor Dragonfly system equipped with a PlanApo objective (63×, 1.4 numerical aperture, oil) and a Zyla scientific complementary metal oxide semiconductor (sCMOS) camera; airyscan imaging was performed using Zeiss LSM 880 Airyscan. Images were analyzed in FIJI (RRID:SCR_002285).

See https://doi.org/10.17504/protocols.io.bvgmn3u6 for details.

### 4Pi-SMS microscopy
Hela cells were seeded on 30-mm diameter No. 1.5H round coverslips (Thorlabs), transfected with VPS13B^GFP, and fixed 24 h after transfection by 4% PFA followed by permeabilization with 0.3% NP40 + 0.05%TX-100. Samples were incubated with mouse anti-GM130 (BD), rabbit anti-TGN46 (Proteintech), chicken anti-GFP (Invitrogen) at 1:500 overnight at 4°C. Samples were washed and were incubated with secondary antibodies goat anti-chicken IgG dylight 650 (Invitrogen), and goat anti-rabbit IgG dylight 633 (Invitrogen) for 2 h, and then with goat anti-mouse IgG CF680 (Biotium) for 1 h at room temperature. After antibody incubation, samples were post-fixed in 3% PFA +0.1% GA (Glutaraldehyde) for 10 min and stored in PBS at 4°C.

Three-color 4Pi-SMS (Zhang et al., 2020) imaging was done on a custom-build microscope with two opposing objectives in 4Pi configuration (Huang et al., 2016). Sample mounting, image acquisition, and data processing were mostly as described before (Zhang et al., 2020) except that imaging speed was 200 Hz with 642 nm laser intensity of 12.5 kW/cm². Typically 100~200 stacks, 3,000 frame/stack were recorded for one cell. DME were

(Drift at Minimum Entropy) used for drift correction (Cnossen et al., 2021). All 4Pi-SMS images and videos were rendered using Point Splatting mode (10-nm particle size) with Vutara SRX 7.0.06 software (Bruker, https://www.bruker.com/en/products-and-solutions/fluorescence-microscopy/super-resolution-microscopes/vutara-vxl.html).

For a detailed protocol please see: https://doi.org/10.1038/s41596-020-00428-7 (Wang et al., 2021).

### FLASH-PAINT
***Imaging buffer.*** Buffer C:1xPBS, 500 mM NaCl. The imaging buffer was supplemented with 1× Trolox, 1× PCA, and 1× PCD (see paragraph below for details).

***Preparation of Trolox, PCA, and PCD.*** 100× Trolox: 100 mg Trolox, 430 μl 100% methanol, 345 μl 1M NaOH in 3.2 ml $H_2O$.

40× PCA: 154 mg PCA, 10 ml water and NaOH were mixed and pH was adjusted 9.0.

100× PCD: 9.3 mg PCD, 13.3 ml of buffer (100 mM Tris-HCl pH 8, 50 mM KCl, 1 mM EDTA, 50 % glycerol). All three were frozen and stored at –20°C.

***Cell fixation preserving Golgi complex.*** Cells were fixed with 4% PFA for 30 min. After four washes (30 s, 60 s, 2 × 5 min), cells were blocked and permeabilized with 3% BSA and 0.25% Triton X-100 at room temperature for 1 h. Next, cells were incubated with the anti-GM130 antibody, anti-COPI antibody, and the GFP-nanobody in 3% BSA and 0.1% Triton X-100 at 4°C overnight. Additionally, all other primary antibodies were preincubated with the corresponding nanobodies (Table S2) at 4°C overnight. The next day, after four washes (30 s, 60 s, 2 × 5 min) cells were incubated with the secondary nanobodies corresponding to GM130 antibody and COPI antibody for ~2 h at room temperature. Next, unlabeled excess secondary nanobodies (to block unlabeled epitopes) were added to preincubation antibody–nanobody mixes at room temperature for 5 min. Next, the antibody–nanobody solutions were pooled and the cells were incubated with the pooled antibody–nanobody mix for ~2.5 h at room temperature. After four washes (30 s, 60 s, 2 × 5 min), the sample was post-fixed with 3% PFA and 0.1% GA for 10 min. Finally, the sample was washed three times with 1× PBS for 5 min each before adding the imaging solution.

***Super-resolution microscope.*** Fluorescence imaging was carried out on an inverted Nikon Eclipse Ti2 microscope (Nikon Instruments) with the Perfect Focus System, attached to an Andor Dragonfly unit. The Dragonfly was used in the BTIRF (Borealis Total Internal Reflection Fluorescence) mode, applying an objective-type TIRF configuration with an oil-immersion objective (Nikon Instruments, Apo SR TIRF 60×, NA 1.49, Oil). As an excitation laser, a 561 nm (1W nominal) was used. The beam was coupled into a multimode fiber going through the Andor Borealis unit reshaping the beam from a Gaussian profile to a homogenous flat top. As a dichroic mirror, a CR-DFLY-DMQD-01 was used. Fluorescence light was spectrally filtered with an emission filter (TR-DFLY-F600-050) and imaged on a sCMOS camera (Sona 4BV6X; Andor Technologies) without further magnification, resulting in an effective pixel size of 108 nm.

***Imaging conditions.*** Imaging was carried out using the corresponding Imager (Tables S3 and S6), Adapter (Tables S4 and S6), and Eraser (Table S5) in imaging buffer. 30,000 frames were acquired at 25-ms exposure time. The readout bandwidth was set to 540 MHz. Laser power (at 561 nm) was set to 80 mW (measured before the back focal plane of the objective), corresponding to ~1.8 kW/cm$^2$ at the sample plane. After imaging, the sample was subsequently washed three times with 200 µl of 1× PBS (on the microscope) followed by an incubation with the corresponding eraser (at 20 nM) for 3 min. This process was repeated iteratively for seven sequential rounds.

***Image processing and analysis.*** DNA-PAINT data was reconstructed, postprocessed (drift correction and alignment of imaging rounds), and rendered with the Picasso package (https://github.com/jungmannlab/picasso.git) (Schnitzbauer et al., 2017). To generate the distance heatmap, we calculated the distance from every localization to its nearest neighbor with respect to every other protein channel, with a distance cutoff at 500 nm. Next, we calculated the median of all determined distances for each protein combination. This median distance is then represented in the heatmap.

More details on FLASH-PAINT can be found at https://doi.org/10.1016/j.cell.2024.02.033 (Schueder et al., 2024).

### Fractionation of FAM177A1-GFP expressing cells

HeLa cells are cultured and transfected as described above. 24 h after transfection, cells are washed three times with ice-cold PBS following lysis with ice-cold fractionation buffer (25 mM Tris pH 7.4, 150 mM NaCl, protease inhibitor). Lysates were ultracentrifuged in a benchtop ultracentrifuge at 50,000 K for 1 h at 4°C, and after centrifugation, the supernatant and pellet were separated. Samples were solubilized in 4X Laemni buffer for western blot analysis.

Please see https://doi.org/10.17504/protocols.io.n2bvjn2ypgk5/v1 for more detail.

### Generation of VPS13B and FAM177A1 KO HeLa cells

sgRNAs targeting the human VPS13B (gRNA1 5′-CAAAATCAT CAATCAAACCG-3′ and gRNA2 5′-AGTGAAAGCTGTAGATCC GA-3′) and FAM177A1 (gRNA 5′-ATATAGATGAGTAACGAA AG-3′) genes were generated using the IDT-DNA and Synthego online tool (https://design.synthego.com/#/, RRID: SCR_024508) HeLa cells were transiently transfected using FuGene HD (Promega) with plasmids containing Cas9 and the sgRNAs (the backbone used was plasmid PX458, a gift from F. Zhang [Broad Institute, Cambridge, MA; RRID:Addgene_48138]). Transfected GFP-expressing cells were then selected by FACS after 48 h via GFP expression, and clonal cell populations were isolated approximately a week after. Mutations in the *FAM177A1KO* (RRID: CVCL_E1HS), *VPS13B1 KO[1]* (RRID: CVCL_E1HQ), *VPS13B KO[2]* (RRID: CVCL_E1HR), and *VPS13BKO[1];FAM177A1 DKO* (RRID: CVCL_E1HT) HeLa cells were confirmed by PCR and sequencing using the primers listed in (Table S1) and by western blot. A detailed description can be found in protocols.io: https://doi.org/10.17504/protocols.io.5jyl85x89l2w/v1.

### BFA treatment

70–80% confluent HeLa cells were treated with 5 µg/ml final BFA (Catalog no #B7651; Sigma-Aldrich) for 1 h at 37°C. BFA was washed out by three brief rinses in PBS and cells were then incubated at 37°C for indicated times before fixation with 4% PFA (catalog no: 15710; EMS). For the quantification of Golgi complex recovery following BFA treatment, the Ilastik (RRID: SCR_015246) 1.4.OS machine-learning software was used (https://github.com/ilastik). The software was trained to recognize compact perinuclear GM130 signal (assembled Golgi complex) versus dispersed GM130 signal (disassembled Golgi) using an independent training dataset. Maximum intensity projections of unprocessed images of the GM130 signal were used.

### Image processing, analysis, and statistics

Statistical analysis was performed with GraphPad Prism 7 software (RRID:SCR_002798). Groups were compared using a two-tail unpaired Student *t* test, and results were deemed significant when the P value was <0.05.

### Zebrafish husbandry

All zebrafish studies reported here were approved by the Institutional Animal Care and Use Committee at Washington University in St. Louis. Zebrafish at all developmental stages were maintained at 28.5°C unless otherwise specified using the standard operating procedures and guidelines established by the Washington University Zebrafish Facility, described in detail at https://zebrafishfacility.wustl.edu/facility-documents/. AB* and *fam177a1a[stl700/stl700]*; *fam177a1b[stl746/stl746]* (*fam177a1a/b DKO*) (Kohler et al., 2024) zebrafish lines were used in this study.

### Annotation of zebrafish vps13b

RNA-seq reads from zebrafish Nadia strain ovary and testis (PRJNA504448, https://doi.org/10.1101/2023.12.06.570431, Wilson and Postlethwait, 2023, *Preprint*) were aligned to the Nadia genome assembly with STAR (RRID:SCR_004463) (Dobin et al., 2013). Sequences were manually annotated from the alignments using Apollo v 2.4.1. (RRID:SCR_001936; https://bio.tools/apollo) (Dunn et al., 2019).

### qRT-PCR

Each RNA sample was isolated using RNeasy Micro Kit (#74004; QIAGEN) from 30 WT embryos. cDNA was synthesized with the iScript kit (#1708841; Bio-Rad) using 1mg of total RNA. qRT-PCR reactions were set up using SsoAdvanced SYBR green (#1725270; Bio-Rad) on CFX Opus 96 (Bio-Rad), and the gene expression was analyzed applying CFX Maestro program (RRID:SCR_016748; Bio-Rad). Primers used are listed in Table S1. A detailed protocol can be seen in the product protocol: https://www.bio-rad.com/sites/default/files/webroot/web/pdf/lsr/literature/10031339.pdf.

### Generation of vps13b crispants
#### gRNA design

To generate *vps13b* crispants, two gRNAs targeting exon 23 of the zebrafish *vps13b* gene were designed using the CHOPCHOP tool (http://chopchop.cbu.uib.no, RRID:SCR_015723 [Montague

et al., 2014]). The target sequences of the gRNAs are 5′-ACT GCCGTCTCCCAGTACGC-3′ and 5′-TCCCGGGCACGGTGCGCA GC-3′. The gRNAs, comprising a duplex made of a target sequence-specific CRISPR RNA and a trans-activating CRISPR RNA, were prepared following the manufacturer's protocol (IDT). RNP mix was assembled with HiFi Cas9 (#1081060; IDT) and ~33 pg of each gRNA and 5 ng of HiFi Cas9 were co-injected into the early one-celled WT and *fam177a1DKO* zygotes. At 25 hpf, we analyzed the overall morphology of the injected embryos, and a randomly selected subset of these embryos was assayed for the target locus disruption through genotyping PCR using *vps13b_* geno_F/R primers (see Table S1) and resolving the resulting PCR products on Fragment Analyzer (Agilent). A detailed protocol can be seen in https://doi.org/10.1038/nbt.2501 (Hwang et al., 2013).

### BFA treatment
BFA (#B7651; Sigma-Aldrich) was dissolved in DMSO and diluted in egg water (60 mg/ml Instant Ocean Sea Salt in distilled water) at 0.8 mg/ml. Zebrafish embryos at 6 hpf were incubated in BFA solution until 25 hpf.

Relevant source data for images and tabular data are deposited to Zenodo: https://doi.org/10.5281/zenodo.11243617.

### Online supplemental material
Fig. S1, related to Fig. 1, further shows the structure and localization of VPS13B. Fig. S2, related to Fig. 2, further shows that human FAM177A1 and FAM177B localize at the Golgi complex. Fig. S3, related to Fig. 3 and Fig. 4, shows the generation of VPS13B and FAM177A1 KO HeLa cells and provides evidence that the Zebrafish genome encodes and expresses a VPS13B ortholog. Table S1 shows the list of primers used in this study. Table S2 shows FLASH-PAINT Nanobody sequences used in this study. Table S3 shows FLASH-PAINT Imager sequences used in this study. Table S4 shows FLASH-PAINT Adapter sequences used in this study. Table S5 shows FLASH-PAINT Eraser sequences used in this study. Table S6 shows FLASH-PAINT Adapter and Imager concentrations used in this study. Table S7 shows a violin plot of localization data represented in Fig. 1 H. Table S8 shows the localization precision table in relation to Fig. 1 H. Table S9 shows the violin plot of localization data presented in Fig. 2 F. Table S10 shows the localization precision table in relation to Fig. 2 F.

### Data availability
The underlying data used in the article are available in Zenodo at https://doi.org/10.5281/zenodo.11243617.

## Acknowledgments
We thank Andrés Guillén-Samander for the helpful discussion.

This work was supported in part by the National Institutes of Health grants R01NS36251 and DA018343 (to P. De Camilli), R01OD011116 (to J. Postlethwait), R01GM151829 (to J. Bewersdorf), and R01HD110556 (to L. Solnica-Krezel), and by the Kavli Institute for Neuroscience (P. De Camilli). This research was also funded in part by Aligning Science Across Parkinson's ASAP-000580 through the Michael J. Fox Foundation for Parkinson's Research.

B. Ugur was a Fellow supported by the William N. and Bernice E. Bumpus Foundation. F. Schueder acknowledges support from the Human Frontier Science Program (LT000056/2020-C). For the purpose of open access, the author has applied a CC BY public copyright license to all Author Accepted Manuscripts arising from this submission.

Author contributions: B. Ugur: Conceptualization, Data curation, Funding acquisition, Investigation, Project administration, Resources, Validation, Visualization, Writing—original draft, Writing—review & editing, F. Schueder: Investigation, Software, Visualization, Writing—original draft, J. Shin: Formal analysis, Resources, M.G. Hanna: Methodology, Y. Wu: Investigation, M. Leonzino: Methodology, Resources, Writing—review & editing, M. Su: Investigation, Methodology, Resources, Visualization, Writing—review & editing, A.R. McAdow: Investigation, C. Wilson: Formal analysis, Investigation, J. Postlethwait: Formal analysis, Funding acquisition, Investigation, Supervision, Visualization, Writing—review & editing, L. Solnica-Krezel: Conceptualization, Formal analysis, Funding acquisition, Project administration, Validation, Writing—original draft, Writing—review & editing, J. Bewersdorf: Funding acquisition, Project administration, Resources, Supervision, Writing—review & editing, P.V. De Camilli: Conceptualization, Funding acquisition, Project administration, Resources, Supervision, Validation, Visualization, Writing—original draft, Writing—review & editing.

Disclosures: J. Bewersdorf reported personal fees from Panluminate Inc. outside the submitted work; in addition, J. Bewersdorf and F. Schueder had a patent to FLASH-PAINT pending. No other disclosures were reported.

Submitted: 5 December 2023

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

JCB

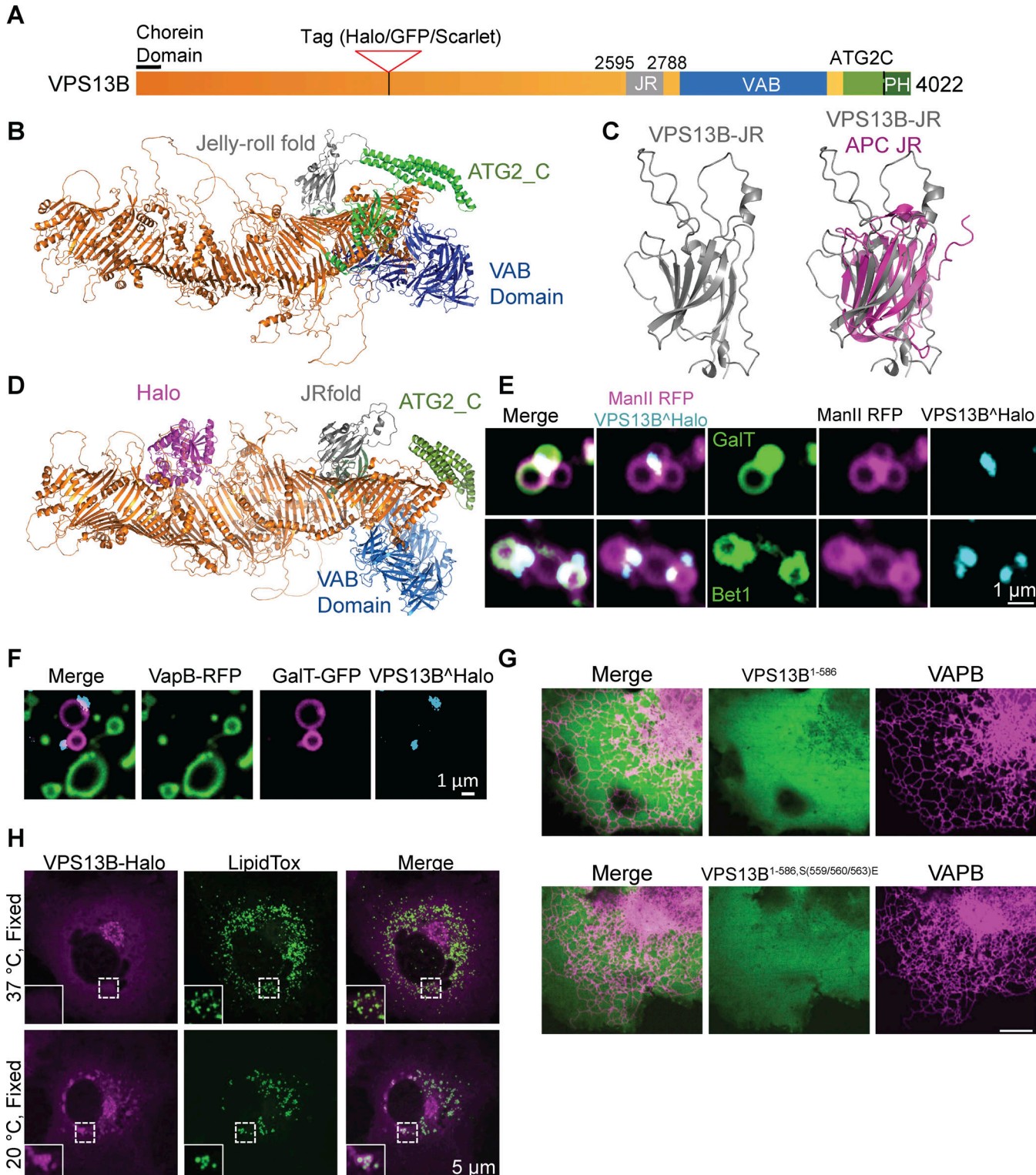

Figure S1. **Structure and localization of VPS13B. (A)** Domain structure of human VPS13B. **(B)** Alphafold2 predicted structure of full-length human VPS13B; orange indicates the rod consisting of 13 RBG domains. **(C)** Alphafold2 prediction of the Jelly Roll (JR) domain of VPS13B alone on the left and overlayed with the jelly-roll fold of anaphase-promoting complex subunit Doc1p/Apc10 on the right. **(D)** Alphafold2 prediction of VPS13B internally tagged with Halo tag (magenta). **(E)** Snapshots of Golgi complex fragments of HeLa cells expressing VPS13B^Halo, ManII-RFP (a medial-Golgi marker), and either GalT-GFP (top panel) or Bet1-GFP (bottom panel) after a 10-min hypotonic shock. Scale bar = 1 µm. **(F)** Snapshots of Golgi complex and ER fragments of HeLa cells expressing VPS13B^Halo, VapB-RFP, and GalT-GFP after a 10-min hypotonic shock. Scale bar = 1 µm. **(G)** Top panel: HeLa cells co-expressing VapB-RFP and the N-terminal fragment of VPS13B (a.a. 1–586). Bottom panel: HeLa cells co-expressing VapB-RFP and the mutant N-terminal fragment (a.a. 1–586): VPS13B^S559E, S560E, S563E. Scale bar = 10 µm. **(H)** COS7 cells expressing VPS13B^Halo and kept at 37°C (top panel) or shifted to 20°C for 30 min before fixation (bottom panel) and then stained with Lipid Tox. Scale bar = 5 µm.

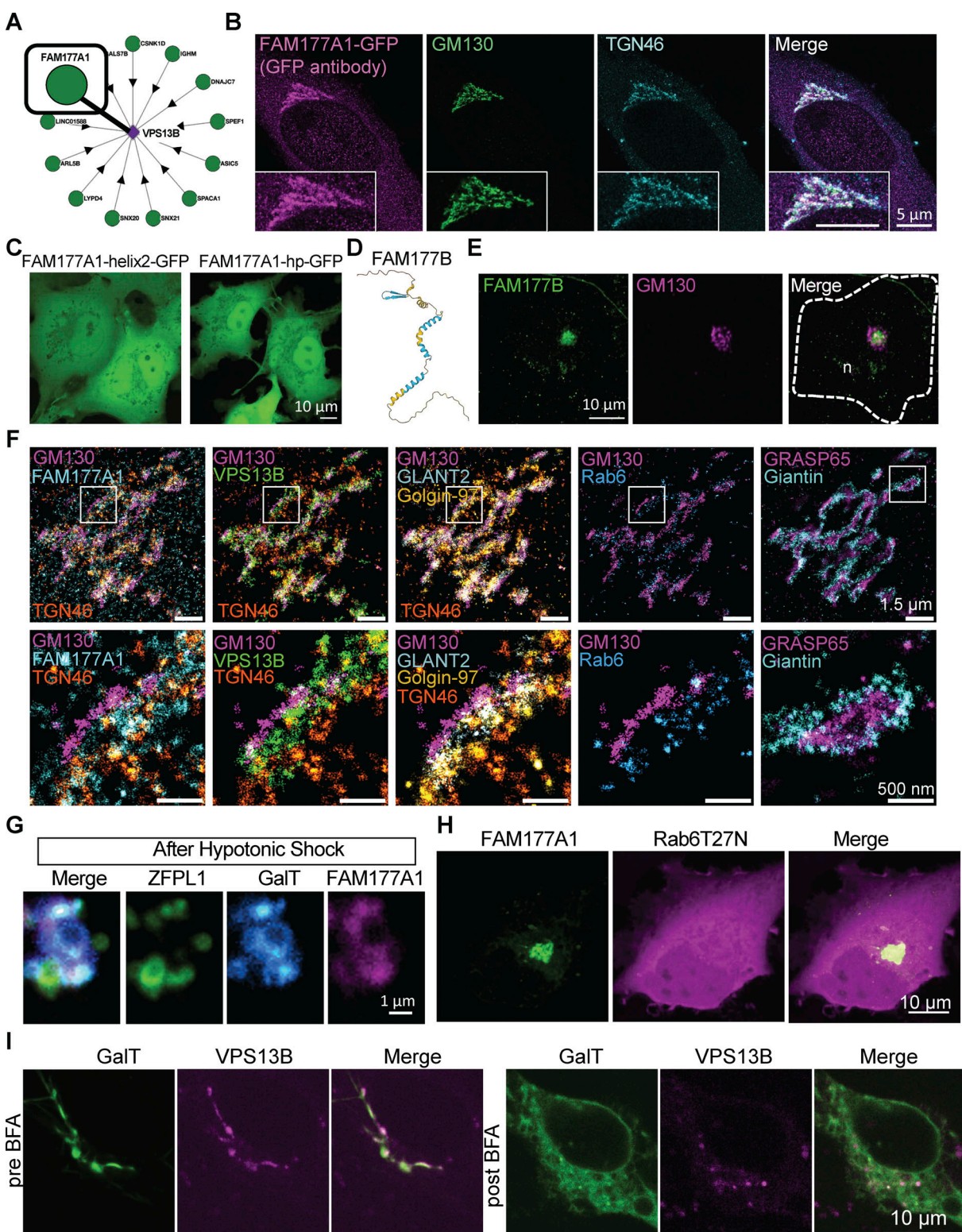

Figure S2. **Human FAM177A1 and FAM177B localize at the Golgi complex. (A)** Bioplex data showing predicted partners of VPS13B (https://bioplex.hms.harvard.edu/explorer/network.php). **(B)** HeLa cells expressing FAM177A1-GFP immunolabeled with anti-GFP, anti-GM130, and anti-TGN46 antibodies. **(C)** COS7 cells expressing FAM177A1-helix2-GFP (left) or FAM177A1-Hairpin-GFP (right). **(D)** Alphafold2 predicted structure of FAM177B. **(E)** HeLa cells expressing FAM177B-flag fixed and immunolabeled with anti-flag and anti-GM130 antibodies. n: nucleus, scale bar = 10 µm. **(F)** FLASH-PAINT performed in HeLa cells expressing VPS13B^GFP and FAM177A1-Halo and immunolabeled with antibodies directed against GFP, halo, GM130, GLANT2, Golgin-97, Rab6, TGN46, GRASP65, and Giantin. **(G)** Snapshots of the Golgi complex of a HeLa cells expressing ZFPL1-GFP, GalT-RFP, and FAM177A1^Halo after a 10-min hypotonic shock. Scale bar = 1 µm. **(H)** HeLa cells expressing FAM177A1-GFP and Rab6T27N-RFP. Scale bar = 10 µm. **(I)** HeLa cells expressing GalT-RFP and VPS13B^Halo before (left panel) and after BFA treatment (5 µg/ml for 40 min, right panel). Scale bar = 10 µm.

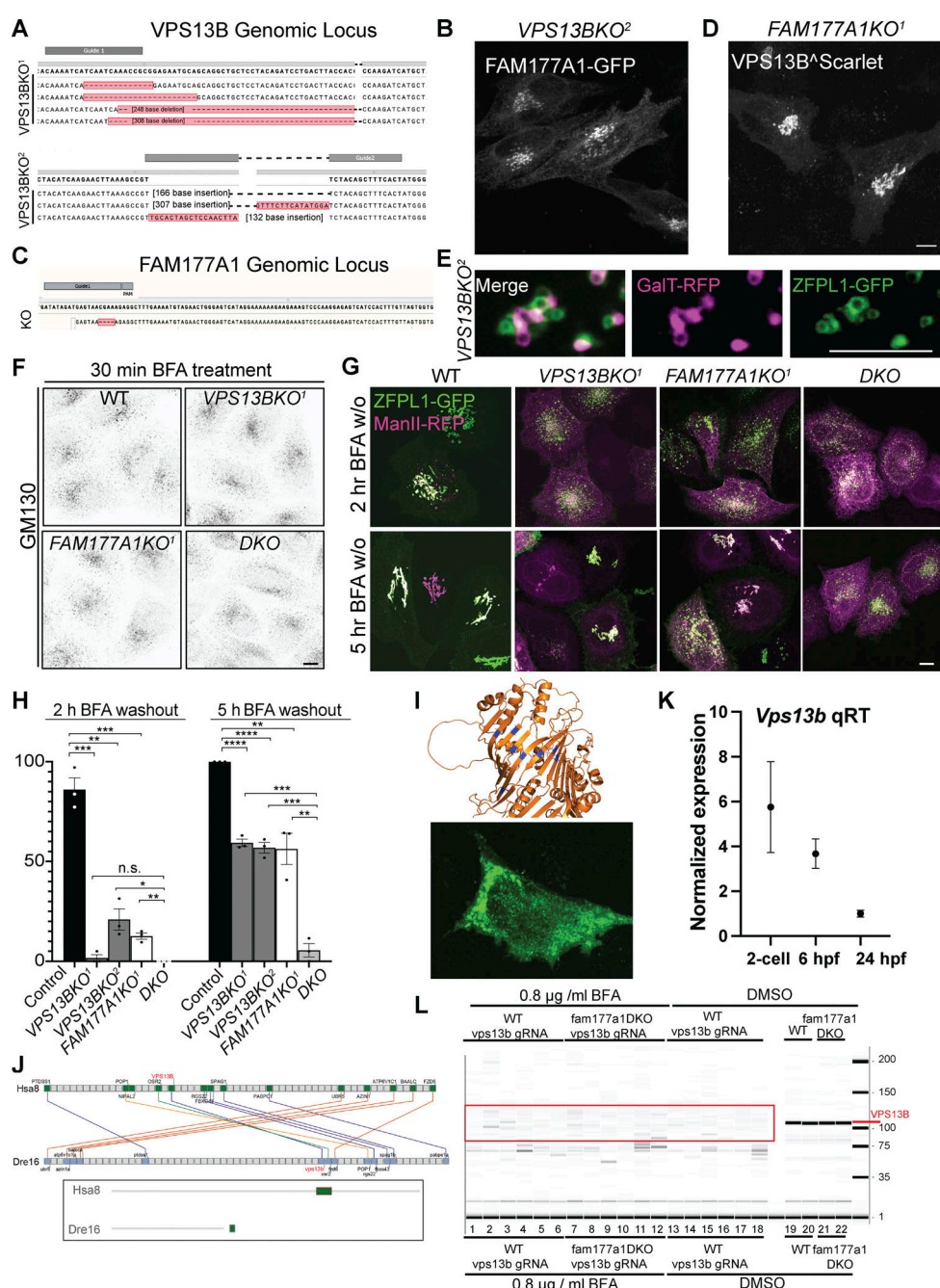

Figure S3. **Generation of VPS13B and FAM177A1 KO HeLa cells and evidence that the zebrafish genome encodes and expresses a VPS13B homolog.** **(A)** Sanger sequencing of *VPS13BKO[1]* and *VPS13BKO[2]* HeLa cells; superscripts indicate different clones. **(B)** *VPS13BKO[2]* HeLa cells expressing FAM177A1-GFP. **(C)** Sanger sequencing of *FAM177A1KO* homozygous HeLa cells. **(D)** *FAM177A1KO* cells expressing VPS13B^Scarlet. Scale bar = 10 μm. **(E)** Snapshots of Golgi fragments of a *VPS13BKO[2]* HeLa cell expressing ZFPL1-GFP, and GalT-RFP after a 10-min hypotonic shock. Scale bar = 5 μm. **(F)** Anti-GM130 immunofluorescence of WT, *VPS13BKO[1]*, *FAM177A1KO*, and *VPS13B;FAM177A1* DKO cells after 30 min in BFA (5 μg/ml). **(G)** WT, *VPS13BKO[1]*, *FAM177A1KO*, and *FAM177A1;VPS13B* DKO *HeLa* cells transfected with ZFPL1-GFP and ManII-RFP after 1 h in BFA (5 μg/ml) followed by subsequent washings as indicated. **(H)** Quantification of Golgi complex reformation in cells of the indicated genotypes after BFA washout for 2 or 5 h. Data are mean ± SEM *n* = 3 per condition; in each condition, 20–50 cells were quantified. Unpaired, two-tailed *t* tests. n.s., not significant. ***P < 0.001; **P < 0.01; *P < 0.05. **(I)** Top: Sites (blue) within the RBG structure of VPS13B where hydrophobic amino acids facing the floor of the hydrophobic grove were replaced by charge amino acids (L65K, I81E, L90E, I155R, L169E, A176E, I203R, L238D, I355K, L264R) to generate the Lipid Transport Dead (LTD) Mut1. Bottom: HeLa cell expressing VPS13[LTDmut1] showing small clusters of the protein sparse throughout the cytoplasm rather than a Golgi localization. **(J)** Conserved syntenies of *VPS13B* in human and zebrafish validate orthology implied by sequence comparisons. A small part of human (*Homo sapiens*) chromosome 8 (Hsa8, green part in insert) has conserved synteny with a short portion of zebrafish (*Danio rerio*) chromosome16 near its right tip (Dre16, green portion in insert). **(K)** qRT-PCR analysis of WT *vps13b* in early zebrafish embryos at two-cell stage, 6 hpf, 24 hpf. **(L)** Genotyping of *vps13b* CRISPR target locus in WT zebrafish embryos injected with *vps13b* gRNAs/Cas9 and treated with 0.8 μg/ml BFA (samples 1–6), *fam177a1;fam177a1b DKO* embryos injected with *vps13b* gRNAs/Cas9 and treated with 0.8 μg/ml BFA (samples 7–12), WT zebrafish embryos injected with *vps13b* gRNAs/Cas9 and treated with DMSO (samples 13–18), WT zebrafish embryos treated with DMSO (samples 19, 20), and *fam177a1;fam177a1b DKO* embryos injected with *vps13b* gRNAs/Cas9 and treated with DMSO (samples 21, 22).

Video 1.  **Rotation of HeLa cells expressing codon-optimized human VPS13B^GFP immunolabeled for GFP (in magenta), GM130 (in yellow), and TGN46 (in green) imaged using 4Pi SMS microscopy and shown in** Fig. 1 C. Scale bar = 1 µm.

Video 2.  **Live imaging of COS7 cell in** Fig. 1 I **expressing VPS13B^halo, ZFPL1-GFP (a cis-Golgi marker), and GalT-RFP (a trans-Golgi marker) during hypotonic shock.** Time scale = min:sec, speed is 7 frames per second.

Video 3.  **Live imaging of HeLa cell in** Fig. 2 H **expressing FAM177A1 GFP and VPS13B^Halo after addition of BFA (5 µg/ml).** Speed is 7 frames per second.

**Provided online are Table S1, Table S2, Table S3, Table S4, Table S5, Table S6, Table S7, Table S8, Table S9, and Table S10. Table S1 shows the list of primers used in this study. Table S2 shows FLASH-PAINT Nanobody sequences. Table S3 shows FLASH-PAINT Imager sequence. Table S4 shows FLASH-PAINT Adapter sequences. Table S5 shows FLASH-PAINT Eraser sequences. Table S6 shows FLASH-PAINT Adapter and Imager concentrations. Table S7 shows a violin plot of data represented in Fig. 1 H. Table S8 shows the localization precision table in relation to Fig. 1 H. Table S9 shows a violin plot of data presented in Fig. 2 F. Table S10 shows the localization precision table in relation to Fig. 2 F.**

