## [Peer Review File · The Journal of Cell Biology]

VPS13B is localized at the interface between Golgi cisternae and is a functional partner of FAM177A1

Berrak Ugur, Florian Schueder, Jimann Shin, Michael Hanna, Yumei Wu, Marianna Leonzino, Maohan Su, Anthony McAdow, Catherine Wilson, John Postlethwait, Lilianna Solnica-Krezel, Joerg Bewersdorf, and Pietro De Camilli

Corresponding Author(s): Pietro De Camilli, Yale School of Medicine and Berrak Ugur, Yale University

Review Timeline:

Submission Date:	2023-12-05
Editorial Decision:	2024-01-23
Revision Received:	2024-05-31
Editorial Decision:	2024-06-24
Revision Received:	2024-07-31

Monitoring Editor: Jodi Nunnari

Scientific Editor: Andrea Marat

Transaction Report:

DOI: <https://doi.org/10.1083/jcb.202311189>

January 23, 2024

Re: JCB manuscript #202311189

Dr. Pietro V De Camilli
Yale School of Medicine
Cell Biology and Neuroscience
100 College Avenue
room 333
New Haven, CT 06510

Dear Dr. De Camilli,

Thank you for submitting your manuscript entitled "VPS13B is localized at the cis-trans Golgi complex interface and is a functional partner of FAM177A1". The manuscript was assessed by expert reviewers, whose comments are appended to this letter. We invite you to submit a revision if you can address the reviewers' key concerns, as outlined here.

You will see that the reviewers appreciate the potential interest in describing a function for VPS13B and FAM177A1 at the Golgi apparatus. However, the reviewers question the localization of VPS13B to an interface between cis and trans Golgi, and we agree that their requested experiments to provide further evidence for the Golgi localization are required. The reviewers also bring up interesting questions regarding the underlying mechanism. For the current study a detailed examination of for example a role for VPS13B in Golgi trafficking of secretory proteins, or potential VPS13B binding partners is not required. However, all suggested experiments with a transport inactive mutant should be attempted barring technical issues such as with folding. In your final revision please also update the editors as to the status of the related papers. Otherwise, we expect you to address all of the remaining reviewer comments in your revised manuscript.

GENERAL GUIDELINES:

Text limits: Character count for a Report is < 20,000, not including spaces. Count includes title page, abstract, introduction, the joint Results & Discussion, and acknowledgments. Count does not include materials and methods, figure legends, references, tables, or supplemental legends.

Figures: Reports may have up to 5 main text figures, if necessary you may expand the study to an Article with 10 figures. To avoid delays in production, figures must be prepared according to the policies outlined in our Instructions to Authors, under Data Presentation, <https://jcb.rupress.org/site/misc/fora.xhtml>. All figures in accepted manuscripts will be screened prior to publication.

*****IMPORTANT:** It is JCB policy that if requested, original data images must be made available. Failure to provide original images upon request will result in unavoidable delays in publication. Please ensure that you have access to all original microscopy and blot data images before submitting your revision. ***

Supplemental information: There are strict limits on the allowable amount of supplemental data. Reports may have up to 3 supplemental figures. Up to 10 supplemental videos or flash animations are allowed. A summary of all supplemental material should appear at the end of the Materials and methods section.

Please note that JCB now requires authors to submit Source Data used to generate figures containing gels and Western blots with all revised manuscripts. This Source Data consists of fully uncropped and unprocessed images for each gel/blot displayed in the main and supplemental figures. Since your paper includes cropped gel and/or blot images, please be sure to provide one Source Data file for each figure that contains gels and/or blots along with your revised manuscript files. File names for Source Data figures should be alphanumeric without any spaces or special characters (i.e., SourceDataF#, where F# refers to the associated main figure number or SourceDataFS# for those associated with Supplementary figures). The lanes of the gels/blots should be labeled as they are in the associated figure, the place where cropping was applied should be marked (with a box), and molecular weight/size standards should be labeled wherever possible.

The typical timeframe for revisions is three to four months. While most universities and institutes have reopened labs and allowed researchers to begin working at nearly pre-pandemic levels, we at JCB realize that the lingering effects of the COVID-19 pandemic may still be impacting some aspects of your work, including the acquisition of equipment and reagents. Therefore, if you anticipate any difficulties in meeting this aforementioned revision time limit, please contact us and we can work with you to find an appropriate time frame for resubmission. Please note that papers are generally considered through only one revision cycle, so any revised manuscript will likely be either accepted or rejected.

Thank you for this interesting contribution to Journal of Cell Biology. You can contact us at the journal office with any questions at cellbio@rockefeller.edu.

Sincerely,

Jodi Nunnari, Ph.D.
Editor-in-Chief

Andrea L. Marat, Ph.D.
Senior Scientific Editor

Journal of Cell Biology

Reviewer #1 (Comments to the Authors (Required)):

The manuscript reports the analysis of the sub-Golgi localization, Golgi targeting, and role of VPS13B in Golgi structure. This is a member of the VPS13 family that, at odds with the other members, does not possess a FFAT domain and, as confirmed in the manuscript, is not targeted to the ER. The Golgi localization of VPS13B (at the cis side of the Golgi) as well as its role in maintaining Golgi structure were known from previous reports.

Here the authors, via the innovative FLASH-PAINT approach and applying hypotonic shock, define the intra-Golgi localization of VPS13B as being between the cis and TGN and envisage a possible role of VPS13B as a tether between cis and trans compartments of the Golgi. In addition, the authors validate the interaction between VPS13B and FAM177A1, analyze the role of this interaction in Golgi targeting of the two proteins, and report the functional interaction of the two proteins in cells and in zebrafish. The data are of high quality and very clearly presented. However, there are some aspects that should be clarified and would benefit from further experimental work.

1. Sub-Golgi localization of VPS13B. Among the different Golgi markers analyzed via the FLASH-PAINT approach, the authors report that the closest one to VPS13B is GM130 (cis). However, from the images presented, the GalT marker appears to be the one that is best colocalizing with VPS13B, both under standard conditions and upon hypotonic shock (Fig.1 G, Fig. 2B). The colocalization with GalT appears even more convincing than the colocalization with the VPS13B interactor FAM177A1 (Fig.1G and 2E). Surprisingly, the distance between GalT and VPS13B is not reported. In addition, to more precisely define the sub-Golgi localization of VPS13B (cis/medial/trans, whether in the compact area of the cisternae, or at the rims or at the intercisternal space), an immunoEM analysis should be performed.

2. Role of VPS13B at the Golgi. The manuscript confirms the reported role of VPS13B in keeping the Golgi structure. What has not been analyzed (with the exception of acrosome biogenesis) is the role of VPS13B in the Golgi trafficking of different secretory proteins. Such an analysis would endow the manuscript with an important element of novelty and should include a study of the role of VPS13B on different types of cargoes to, through, and out of the Golgi.

3. Mechanism of action of VPS13B at the Golgi. The authors propose that VPS13B might act as a tether between the cis and trans Golgi compartments. However, the length of VPS13B (max length around 30 nm) would not fit with the average distance between the cis and trans compartments of the Golgi, which should reach on average 200 nm or more (considering an average of six cisternae/stack). The authors should explain how their model would fit with the actual width of Golgi stacks.

Of course, a very interesting scenario would be one in which VPS13B acts as an inter-cisternae lipid transporter. The manuscript would benefit greatly from an experimental dissection between these two possible mechanisms of action (using, if available, VPS13B mutants which are targeted to the Golgi but are inactive in lipid transport) or at least from a discussion on this aspect based on the analysis of the available data on VPS13B mutants (reported in Zorn et al, 2022).

Minor comments

Fig. 2F: Post BFA the panels of single and merge staining are misplaced.

Fig. 3 Have the authors followed the recovery of other Golgi markers (TGN or enzymes) after the BFA wash-out?

Reviewer #2 (Comments to the Authors (Required)):

This manuscript presents novel findings on the localization and role of VPS13B at the Golgi apparatus. Interestingly, the authors also report on a VPS13B interacting protein, FAM177A1, which appears to share some functions with VPS13B. Super-resolution microscopy (FLASH-PAINT) data has been exploited to report on the relative distances between VPS13B and a number of Golgi markers. This, together with hypotonic shock treatment-based experiments, prompted the authors to suggest that VPS13B (and FAM177A1) localize at cis-Golgi-trans-Golgi contacts. They later found that these two proteins are required for a fast reformation of the Golgi after BFA-induced disassembly.

Although the paper is generally interesting and presents potentially important and novel findings, I have a list of concerns that prevent me at the moment of recommending this manuscript for publication at the JCB:

Major comments:

- Line 136 (and Fig. 1): the authors performed an "analysis of distances of the various antigens from each other", (maybe I missed it) but I was not able to find the methodology of how this analysis was performed from the localization microscopy data. Without this information I am not sure I can judge properly about the validity of this analytical approach.
- Along the same lines, in Fig. 1F, the median distances are reported. Could the authors show full histograms of the distances (as there could be different populations). What is the rationale to use a "cut off distance 500 nm"?
- Can the authors report the obtained localization precision/accuracy from the FLASH-PAINT experiments?
- Are the results shown in Fig. 1G specific for the chosen set of cis and trans Golgi markers? Have they tested different markers (as e.g., any of the ones used in Fig. 1F)? Out of curiosity, have the authors tested the effect of a hypotonic shock on the formation of these cis/trans contacts in VPS13B KO cells? (i.e., is VPS13B somewhat necessary for cis/trans contacts?)
- Fig. 2B: I think it would be very informative to obtain the relative localization of VPS13B and FAM177A1 by FLASH-PAINT. Are the distances between these two proteins shorter than for any pairs? Is FAM177A1 more cis or trans localized? or equally distant/proximal to both (as suggested by results in Fig. 2E)?
- Line 271: as the authors claim synergy between the proteins, have they tested whether the phenotypes of the single KOs can be rescued by the overexpression of the complementary protein (e.g., VPS13B KO + o/expr FAM177A1)?
- It is not clear to me what the zebrafish experiments are adding to the story.
- The authors showed that the dynamics of Golgi reassembly after BFA treatment is dependent on having functional VPS13B or FAM177A1. Although they suggest some possible explanations for this (e.g., lipid transfer defects), this appears as merely an observation at this point, with no clear mechanism. Is VPS13B-mediated lipid transfer necessary for this to happen?
- If VPS13B is at cis-trans MCS, what are the binding partners on each membrane? Rab6 has been suggested to be a trans-binder, but what about the cis side?
- Regarding the similarity between the phenotypes in mammalian cells and in zebrafish, would the authors expect (and could they test) whether the zebrafish vps13b overexpression in VPS13B KO HeLa cells could rescue, at least partially, the phenotypes? Or still they are quite different proteins?
- In summary, I am not convinced that the data shown here really reports that VPS13B localizes at contacts between cis and trans Golgi membranes, nor what the function of this protein is in the events leading to Golgi reassembly after BFA treatment.

Minor comments:

- Line 99: This is also shown in Fig. 1A.
- Line 157: I'd show these data as a supporting figure.
- Line 200: "GAPDH"
- Line 246: This is probably not so important, but I'm not sure if GM130 is the best marker for BFA experiments, as it is a peripheral protein, and dispersal could be due to relocation to ER/cytosol or due to the canonical BFA effect. A transmembrane protein (e.g. Mannosidase II) would be better.
- Line 250-251: "Treatment of VPS13B KO cells with BFA induced dispersion of the Golgi complex with a time course similar to control cells". The authors used only a single time for the BFA treatment (rather long actually), so no conclusions of the dispersion dynamics can be drawn from these data.

Reviewer #3 (Comments to the Authors (Required)):

This is a high quality study of the role of VPS13B in Golgi architecture and overall function, together with analysis of FAM177A1, a genetic interactor. Apart from studying expressed constructs, the authors have used several other approaches to study these two proteins, many of which have extreme difficulty and produce great sensitivity. In addition to working with human cells, they have also worked in embryonic development of fish.

Overall, the work is excellent and will make a fine publication for JCB.

I have one major issue, which is ascribing the localization of VPS13B to an interface between cis and trans Golgi compartments. Hypotonic shock to expand Golgi cisternae (Fig 1G) is ingenious, but the results are confusing: there are several circular profiles that are positive for both ZFP1-1 and GalT, and VPS13B tends to be in irregular patches located near to these sites of cis-trans overlap and NOT specifically enriched in regions that are between ZFP1-1 single-positive profiles and GalT single-positive profiles. Although it is not clear what the overlap of markers here means, various statements all seem wrong ["VPS13B is localized at the interface between cis and trans Golgi sub-compartment " (line 29 & 75); "closely apposed, but distinct vacuoles" (line 153); "VPS13B selectively localized at the interface of the two types of vacuoles" (line 154); "Overall, our results suggest that in contrast to other VPS13 proteins, VPS13B may bridge cis and trans Golgi membranes." (Line 171); "VPS13B...was restricted to the cis-trans Golgi interface (Fig. 2E)" (line 223); "VPS13B coalesces at contacts between large vesicles positive for cis-Golgi and trans-Golgi markers " (line 333)]. Could the VPS13B+ve regions be pools of tubulovesicular elements with mixed origin from several compartments?

The problem with the explanation given for VPS13B function "transfer of phospholipids from cis- to trans-Golgi membranes via a bridge-like mechanism" (line 337) is not that it is highly speculative, but rather that it makes no sense. After BfA, TGN might reform from EE, and CGN might form from cargo, but how do other elements form? After stack formation, as I see it there is no possible contact between cisternae at opposite sides of the stack - a medial cisterna would intervene. Also the bridge can only be [less than or equal to]27 nm. Could VPS13B be on a ("the") medial cisterna, maybe binding on both sides of this, thus bridging to both cis and trans?

Other points:

Fam177A1 conservation: the 1st paragraph on this protein (p5) would be a good place to describe the full range of organisms that have a homologue: this includes a very wide range of eukaryotes, as shown either from IPR028260. Alignments are informative of the regions that are likely to interact with the conserved VPS13B partner (e.g. in *Capsaspora* protein CAOG_09126: aligns best with residues 93-154, including H1 and ~70% of H2, the 1st 12 aa of H2 (KYQYAIDEYYRM) forming a highly conserved amphipathic helix. HHpred additionally finds previously unknown homologues more widely in plants (e.g. At2g18910), which is interesting as *A. thaliana* has no obvious VPS13B ortholog.

Fam177A1 constructs: I could not find any co-ordinates of these constructs. These are important for others to reproduce the work. Another aspect to consider is how to subdivide the protein rationally. For example, the residues between predicted H1 and H2 126GIST129 are among the most conserved in the whole protein - are they involved with H1's targeting function, or with the function H2 (whatever that is)?

Fam177A localization (Fig 2E): FAM177A1 is membrane bound and it has a wider distribution than both ZPFL-1 and VPS13B. It would be good to know if the FAM177-positive ZPFL1-negative circular profiles are positive for another Golgi marker.

Minor:

what is the article Legro et al. (in revision)? Can we be told more - at least a title and author list (in a footnote)? Unlike this MS I was unable to find that one in a preprint server.

Fig 1C-E: the text should explain whether VPS13B is more punctate than the other markers, and also compare the extent to which label is found outside the Golgi ribbon

Explain how Fig 1F is not symmetrical X->Y and Y->X

Fig 1G has no scale bar.

It would help to show the experiments with VPS13B and ER/Golgi markers (line 157).

Can Fam177 binding to VPS13B be studied in silico by co-folding all of it (or the likely region - H2) with segments of VPS13B?

Fig 2F bottom row - images wrongly arranged

Fig. S2G: It is not the full picture that "GalT did not colocalize with the remaining VPS13B-positive spots and instead dispersed into the ER as anticipated" (line 239). The few GalT puncta that remain do tend to be at the VPS13B spots.

"VPS13B ortholog had not been annotated in zebrafish" (line 277) - though the MS could add that a 3963 aa orthologue has been identified in *Cyprinus carpio* (Common carp) that is over 80% identical to *D rerio* Vps13b.

Manner of reporting fish results: "indicating a functional interaction" (line 302). The MS could go further: the data show a synthetic phenotype, which aligns with results in Fig 3 and normally indicates a parallel pathway, NOT that the gene products act together as one complex.

V minor:

what is the rule by which sometimes Vps13b is written in place of VPS13B?

1st Revision - Authors' Response to Reviewers: May 31, 2024

J. Cell Biology manuscript #202311189

Title: VPS13B is localized at the interface between Golgi cisterns and is a functional partner of FAM177A1

Authors: Ugur et al.

Response to the reviewers' comments.

We thank the reviewers for their careful reading of the manuscript and insightful comments and suggestions. We have now addressed all their comments with either new experiments or discussion. Several new panels/figures have been added to the manuscript:

Main Fig. 1C, 2E, 2F

Supplemental Figures S1E, S1F, S1G, S2F, S2G, S3E, S3F, S3G, S3H, S3I

Supplemental Tables S7, S8, S9 and S10.

We highlighted major text edits with blue text color. A detailed point-to-point response to their comments is appended below.

We also made a minor modification of the title to avoid a confusion that may have arisen from the original title. The original title was: "VPS13B is localized at the cis-trans Golgi complex interface and is a functional partner of FAM177A1". The reviewers raised the issue that the distance between cis- and trans- Golgi cisterns is incompatible with the length of VPS13B. However, we did not mean to imply that VPS13B directly bridges cis- and trans-Golgi cisterns in situ. Thus, we modified the title that now reads "VPS13B is localized at the interface between Golgi cisterns and is a functional partner of FAM177A1"

Reviewer #1 (Comments to the Authors (Required)):

The manuscript reports the analysis of the sub-Golgi localization, Golgi targeting, and role of VPS13B in Golgi structure. This is a member of the VPS13 family that, at odds with the other members, does not possess a FFAT domain and, as confirmed in the manuscript, is not targeted to the ER. The Golgi localization of VPS13B (at the cis side of the Golgi) as well as its role in maintaining Golgi structure were known from previous reports.

Here the authors, via the innovative FLASH-PAINT approach and applying hypotonic shock, define the intra-Golgi localization of VPS13B as being between the cis and TGN and envisage a possible role of VPS13B as a tether between cis and trans compartments of the Golgi. In addition, the authors validate the interaction between VPS13B and FAM177A1, analyze the role of this interaction in Golgi targeting of the two proteins, and report the functional interaction of the two proteins in cells and in zebrafish. The data are of high quality and very clearly presented.

We thank the reviewer for their very positive comments.

However, there are some aspects that should be clarified and would benefit from further experimental work.

1. Sub-Golgi localization of VPS13B. Among the different Golgi markers analyzed via the FLASH-PAINT approach, the authors report that the closest one to VPS13B is GM130 (cis). However, from the images presented, the GalT marker appears to be the one that is best colocalizing with VPS13B, both under standard conditions and upon hypotonic shock (Fig. 1G, *now Fig. 1I* Fig. 2B). The colocalization with GalT appears even more convincing than the colocalization with the VPS13B interactor FAM177A1 (Fig. 1G and 2E *now Fig. 1I and 2G respectively*). Surprisingly, the distance between GalT and VPS13B is not reported. In addition, to more precisely define the sub-Golgi localization of VPS13B (cis/medial/trans, whether in the compact area of the cisternae, or at the rims or at the intercisternal space), an immunoEM analysis should be performed.

We thank the reviewer for these comments, which we have addressed with new experiments. The GalT construct we have used for our figures is a fragment of the Homo sapiens beta-1,4-galactosyltransferase 1 which is commonly used as a trans-Golgi marker (PMID: 6121819; 7615680; 17021253). In the hypotonic shock images (Fig 1G) one can appreciate the greater localization of VPS13B with ZFPL1 (cis Golgi marker) than with GalT. Concerning the point that we do not report the distance between GalT and VPS13B by FLASH-Paint, this is because for the FLASH-Paint experiments we prefer to use, when possible, antibody labeling to avoid possible artifacts of overexpression (we do this when well validated antibodies

are available). Therefore, we prefer not to overexpress GalT as a trans-Golgi marker and instead use anti trans-Golgi complex antibodies: antibodies directed against TGN46 and Golgin97.

We have now carried out new FLASH-Paint experiments using an antibody directed against GALNT2 (N-Acetyl-Galactoseaminyl Transferase 2, PMID: 36007678), an additional medial Golgi marker, and the VPS13B signal was calculated to be closer to the medial Golgi than to the trans-Golgi (Fig. 2F and S2F). In addition, we carried out 4Pi-single-molecule switching (SMS) microscopy (PMID:33328610). This technique, which provide better isotropic resolution than FLASH-PAINT, but is limited to a maximum of three markers, clearly demonstrates that VPS13B is localized in the space comprised between the cis- and the trans-Golgi (Fig. 1C, Video1). We do not imply that VPS13B directly bridges cis-membranes to trans-membranes, but that VPS13B may perform a bridging function between two membranes in the space comprised between cis- and trans-Golgi that includes medial Golgi membranes. We have attempted immunoEM and CLEM but the results have not been reliable. In view of the information afforded by the superresolution microscopy technique used, we hope that you will consider the manuscript acceptable even without EM.

2. Role of VPS13B at the Golgi. The manuscript confirms the reported role of VPS13B in keeping the Golgi structure. What has not been analyzed (with the exception of acrosome biogenesis) is the role of VPS13B in the Golgi trafficking of different secretory proteins. Such an analysis would endow the manuscript with an important element of novelty and should include a study of the role of VPS13B on different types of cargoes to, through, and out of the Golgi.

We performed RUSH experiments with two cargo proteins that traffic from the Golgi; Cathepsin-D (CatD, a lysosome directed cargo) and LysozymeC (LyzC, a plasma membrane directed cargo). We did not find any obvious difference in their transit through the Golgi complex, as shown in the figures appended below for the reviewers. We cannot exclude that differences may be observed with other cargoes and with other cell types.

Fig. 1. (A) GalT-RFP and LyzC-RUSH (Retention Using Selective Hooks) transfected WT and VPS13BKO HeLa cells were treated with 40 μ M Biotin at indicated time points. (B) LyzC fluorescence intensity in the Golgi complex region at indicated time points was normalized to the maximum value, error bars indicate S.D., n = 3 experiments. (C) GalT-GFP and CatD-RUSH transfected WT and VPS13BKO HeLa cells were treated with 40 μ M Biotin at indicated time points. (D) CatD fluorescence intensity in the Golgi complex region at indicated time points was normalized to the maximum value, error bars indicate S.D., n = 3 experiments. Scale bar = 10 μ m. LyzC-RUSH and CatD-RUSH constructs were kind gifts from von Blume lab.

3. Mechanism of action of VPS13B at the Golgi. The authors propose that VPS13B might act as a tether between the cis and trans Golgi compartments. However, the length of VPS13B (max length around 30 nm) would not fit with the average distance between the cis and trans compartments of the Golgi, which should reach on average 200 nm or more (considering an average of six cisternae/stack). The authors should explain how their model would fit with the actual width of Golgi stacks. Of course, a very interesting scenario would be one in which VPS13B acts as an inter-cisternae lipid transporter.

Collectively, FLASH-Paint imaging, hypotonic shock and the new 4Pi-single-molecule switching (SMS) microscopy data indicate that VPS13B is localized within the space comprised between cis- and trans-regions Golgi, with greater proximity to the cis-Golgi region. As we stated above, we do not imply that VPS13B bridges the cis-Golgi membranes to trans-Golgi membranes directly, but that it bridges adjacent cisterns in the cis-medial Golgi region. Upon hypotonic shock, cis-medial cisterns and medial- trans cisterns coalesce into distinct vacuoles and VPS13B accumulates at the interface of these two sets of vacuoles. What seems to be clear is that VPS13B selectively accumulates at contacts between two Golgi elements with different properties. Based on a published cryo EM structure of the Golgi complex (below), distances between cisterns can be in the range of 25-30nm. We have added orange 25nm rods to Figure 1A from Bykov et al., 2017 so that the distances between cisterns are visible. One should also consider that VPS13B may be tilted and not arranged perpendicularly between apposed cisterns.

Fig. 2. Architecture of the Chlamydomonas Golgi apparatus and transport vesicles revealed by in situ cryo-ET. From Bykov et al., 2017 PMID: 29148969 scale bar: 200 nm. **Orange rods were added by us to represent 25 nm distance between cisterns.**

The manuscript would benefit greatly from an experimental dissection between these two possible mechanisms of action (using, if available, VPS13B mutants which are targeted to the Golgi but are inactive in lipid transport) or at least from a discussion on this aspect based on the analysis of the available data on VPS13B mutants (reported in Zorn et al, 2022).

This is a very important point, which we tried to address with new experiments. We generated a lipid transport dead mutant of human VPS13B (L65K, I81E, L90E, I155R, L169E, A176E, I203R, L238D, I355K, L264R named as LTD Mut1) based

on the one generated in *S. cerevisiae* (L64K/I80E/L87E/I162R/L185E/A192E/L217R/V269E/L275D/M293K/L300R (Li et al., 2020 PMID: 32182622)) by overlaying AlphaFold predicted structures (Fig. S3I). Unfortunately, this construct formed aggregates in cells. Thus, we did not attempt to rescue the BFA phenotype with this construct. This is now stated and shown in the revised manuscript.

We also carefully considered the mutations discussed in the paper by Zorn et al. 2002. Among the 10 patient mutations tested, 6 abolished Golgi complex localization. The remaining 4 localized to the Golgi complex and rescued the mild phenotype that they had described in VPS13B KO cells and were hence defined in the Zorn et al. paper as “variants of known significance”. In fact, two of 4 mutations (Ala590Thr and Ser824Ala) were found at a homozygous state in healthy individuals.

Minor comments

Fig. 2F (*now Fig. 1H*): Post BFA the panels of single and merge staining are misplaced.

Thank you, fixed.

Fig. 3 Have the authors followed the recovery of other Golgi markers (TGN or enzymes) after the BFA wash-out?

We have now performed new experiments with Mannosidase II (Man II) and ZFPL1 (cis Golgi membrane marker) and obtained the similar results (Fig. S3E and S3F.)

Reviewer #2 (Comments to the Authors (Required)):

This manuscript presents novel findings on the localization and role of VPS13B at the Golgi apparatus. Interestingly, the authors also report on a VPS13B interacting protein, FAM177A1, which appears to share some functions with VPS13B. Super-resolution microscopy (FLASH-PAINT) data has been exploited to report on the relative distances between VPS13B and a number of Golgi markers. This, together with hypotonic shock treatment-based experiments, prompted the authors to suggest that VPS13B (and FAM177A1) localize at cis-Golgi-trans-Golgi contacts. They later found that these two proteins are required for a fast reformation of the Golgi after BFA-induced disassembly.

Although the paper is generally interesting and presents potentially important and novel findings, I have a list of concerns that prevent me at the moment of recommending this manuscript for publication at the JCB:

Major comments:

- Line 136 (and Fig. 1): the authors performed an "analysis of distances of the various antigens from each other", (maybe I missed it) but I was not able to find the methodology of how this analysis was performed from the localization microscopy data. Without this information I am not sure I can judge properly about the validity of this analytical approach.

To generate the distance heatmap, we calculated the distance from every localization to its nearest neighbor with respect to every other protein channel, with a distance cutoff at 500 nm. Next, we calculated the median of all determined distances for each protein combination. This median distance is then represented in the heatmap. This has now been added to “Methods”.

- Along the same lines, in Fig. 1F, *now Fig. 1H*, the median distances are reported. Could the authors show full histograms of the distances (as there could be different populations). What is the rationale to use a "cut off distance 500 nm"?

We have now added as a supplemental table (Supplemental Table 7 and Supplemental Table 8) a matrix of violin plots showing every individual distance distribution calculated as described above. As the reviewer can see by looking at the individual violin plots, all distributions return to zero before the cutoff distance of 500 nm. We are using the 500 nm cutoff to eliminate outlier distances (e.g., localizations from nonspecific binding events or proteins spatially located outside of the Golgi) which might bias the results.

- Can the authors report the obtained localization precision/accuracy from the FLASH-PAINT experiments?

We now report the localization precision of every round of imaging in Supplementary Table S8 and S10.

- Are the results shown in Fig. 1G (*now Fig. 1I*) specific for the chosen set of cis and trans Golgi markers? Have they tested different markers (as e.g., any of the ones used in Fig. 1F, *now Fig. 1H*)?

The chosen cis- and trans- Golgi markers (ZFPL1 and GalT) display a striking segregation in two distinct set of vacuoles generated by the hypotonic shock. We have now performed hypotonic experiments with Mannosidase II (medial), Bet1 (cis-Golgi associated protein) and GalT and the results have confirmed that upon hypotonic shock cis- and trans- Golgi markers coalesce into two distinct set of vacuoles and that VPS13B is localized at contact between these two sets of vacuoles. They also show presence of the medial markers in both vacuoles. What seems to be clear is that VPS13B selectively accumulates at contacts between two Golgi elements with different properties. The corresponding data are now in Fig. S1E.

Out of curiosity, have the authors tested the effect of a hypotonic shock on the formation of these cis/trans contacts in VPS13B KO cells? (i.e., is VPS13B somewhat necessary for cis/trans contacts?)

We have now done this experiment and shown in Fig. S3E that cis- to trans- Golgi vacuoles are tethered to each other also in the absence of VPS13B. This not surprising as clearly the structural organization of the Golgi complex requires multiple tethering proteins.

- Fig. 2B: I think it would be very informative to obtain the relative localization of VPS13B and FAM177A1 by FLASH-PAINT. Are the distances between these two proteins shorter than for any pairs? Is FAM177A1 more cis or trans localized? or equally distant/proximal to both (as suggested by results in Fig. 2E *now Fig. 1G*)?

We have now carried out FLASH-PAINT imaging of FAM177A1 relative to 7 Golgi complex markers and found that FAM177A1 has a broad distribution throughout the Golgi complex. These new data are now shown in Fig. 2E and 2F, S3F, Supplemental Tables S9 and S10.

- Line 271: as the authors claim synergy between the proteins, have they tested whether the phenotypes of the single KOs can be rescued by the overexpression of the complementary protein (e.g., VPS13B KO + o/expr FAM177A1)?

This is an interesting suggestion. However, in prioritizing the experiments to carry out for this revision we have decided to not prioritize this experiment as FAM177A1 does not have a structure predicting a lipid transport function and thus is unlikely to rescue the absence of VPS13B. We are willing to perform this experiment if the reviewers deemed this experiment as strictly necessary.

- It is not clear to me what the zebrafish experiments are adding to the story.

We prefer to keep them as they provide evidence for evolutionary conservation pointing to the importance of the partnership of these two proteins.

- The authors showed that the dynamics of Golgi reassembly after BFA treatment is dependent on having functional VPS13B or FAM177A1. Although they suggest some possible explanations for this (e.g., lipid transfer defects), this appears as merely an observation at this point, with no clear mechanism. Is VPS13B-mediated lipid transfer necessary for this to happen?

*This is a very important point that has also been raised by reviewer #1. We tried to address with new experiments. We generated a lipid transport dead mutant of human VPS13B (L65K, I81E, L90E, I155R, L169E, A176E, I203R, L238D, I355K, L264R named as LTD Mut1) based on the one generated in *S. cerevisiae* (L64K/I80E/L87E/I162R/L185E/A192E/L217R/V269E/L275D/M293K/L300R (Li et al., 2020 PMID: 32182622)) by*

overlaying AlphaFold predicted structures. Unfortunately, this construct formed aggregates in cells (Fig. S3I). Thus, we did not attempt to rescue the BFA phenotype with this construct. This is now stated and shown in the revised manuscript.

-If VPS13B is at cis-trans MCS, what are the binding partners on each membrane? Rab6 has been suggested to be a trans-binder, but what about the cis side?

We do not have an answer for this question. We have expressed N-terminal fragment of VPS13B, and this construct has a cytosolic localization (Fig. S1G). This is in contrast to the ER localization of VPS13A, C and D which have FFAT motifs. Possibly, a low affinity interaction with a Golgi complex membrane is sufficient to allow binding when the protein is concentrated there by an interaction with Rab6. We hope to have an answer from future studies.

- Regarding the similarity between the phenotypes in mammalian cells and in zebrafish, would the authors expect (and could they test) whether the zebrafish vps13b overexpression in VPS13B KO HeLa cells could rescue, at least partially, the phenotypes? Or still they are quite different proteins?

In summary, I am not convinced that the data shown here really reports that VPS13B localizes at contacts between cis and trans Golgi membranes, nor what the function of this protein is in the events leading to Golgi reassembly after BFA treatment.

As VPS13B is a very large protein that does not express well, we would need to synthesize a mammalian codon optimized version of the zebrafish gene. The protein similarity between these two proteins is substantial especially at the structural level (see Fig. 3A) and it plausible that the zebrafish protein may rescue the phenotype.

Minor comments:

- Line 99: This is also shown in Fig. 1A.

Now added to the text.

- Line 157: I'd show these data as a supporting figure.

We have now provided new data that show co-expression of N-terminal fragments of VPS13B with VAPB and failed to observe any ER localization of VPS13B (Fig. S1G).

- Line 200: "GAPDH"

Corrected.

- Line 246: This is probably not so important, but I'm not sure if GM130 is the best marker for BFA experiments, as it is a peripheral protein, and dispersal could be due to relocation to ER/cytosol or due to the canonical BFA effect. A transmembrane protein (e.g. Mannosidase II) would be better.

We have now added experiments that show BFA recovery with Mannosidase II (Man II) and ZFPL1 (cis Golgi complex membrane marker) (Fig. S3G and S3H)

- Line 250-251: "Treatment of VPS13B KO cells with BFA induced dispersion of the Golgi complex with a time course similar to control cells". The authors used only a single time for the BFA treatment (rather long actually), so no conclusions of the dispersion dynamics can be drawn from these data.

We have tested Golgi complex dispersal at 15 min and 30 min. In our WT HeLa cell line the Golgi complex is typically dispersed in 15 min and we see a complete dispersal of all cells by 30 min. By 15 min we observe that Golgi complex is also dispersed in VPS13B KO cells. We treat cells with BFA 1hr before recovery to make sure that all the cells have dispersed Golgi complex. We have added a figure (Fig. S3F) that shows dispersal of Golgi complex in WT, VPS13BKO, FAM177A1KO and DKO cells when they are treated with BFA for 30 min.

Reviewer #3 (Comments to the Authors (Required)):

This is a high quality study of the role of VPS13B in Golgi architecture and overall function, together with analysis of FAM177A1, a genetic interactor. Apart from studying expressed constructs, the authors have used several other approaches to study these two proteins, many of which have extreme difficulty and produce great sensitivity. In addition to working with human cells, they have also worked in embryonic development of fish.

Overall, the work is excellent and will make a fine publication for JCB.

We thank this reviewer for his/her positive comments

I have one major issue, which is ascribing the localization of VPS13B to an interface between cis and trans Golgi compartments. Hypotonic shock to expand Golgi cisternae (Fig 1G) is ingenious, but the results are confusing: there are several circular profiles that are positive for both ZFPL-1 and GalT, and VPS13B tends to be in irregular patches located near to these sites of cis-trans overlap and NOT specifically enriched in regions that are between ZFPL-1 single-positive profiles and GalT single-positive profiles. Although it is not clear what the overlap of markers here means, various statements all seem wrong ["VPS13B is localized at the interface between cis and trans Golgi sub-compartment " (line 29 & 75); "closely apposed, but distinct vacuoles" (line 153); "VPS13B selectively localized at the interface of the two types of vacuoles" (line 154); "Overall, our results suggest that in contrast to other VPS13 proteins, VPS13B may bridge cis and trans Golgi membranes." (Line 171); "VPS13B...was restricted to the cis-trans Golgi interface (Fig. 2E)" (line 223); "VPS13B coalesces at contacts between large vesicles positive for cis-Golgi and trans-Golgi markers " (line 333)]. Could the VPS13B+ve regions be pools of tubulovesicular elements with mixed origin from several compartments? The problem with the explanation given for VPS13B function "transfer of phospholipids from cis- to trans-Golgi membranes via a bridge-like mechanism" (line 337) is not that it is highly speculative, but rather that it makes no sense. After BfA, TGN might reform from EE, and CGN might form from cargo, but how do other elements form? After stack formation, as I see it there is no possible contact between cisternae at opposite sides of the stack - a medial cisterna would intervene. Also the bridge can only be {less than or equal to}27 nm. Could VPS13B be on a ("the") medial cisterna, maybe binding on both sides of this, thus bridging to both cis and trans?

We appreciate these comments and realize that our text may have not been clear. Moreover, we have performed experiments with a marker of the medial Golgi (Man II) and with another protein besides ZFPL concentrated in the cis-Golgi (Bet1), and these experiments have helped us to better understand the results of the hypotonic shock experiments.

Definitely, we do not imply that the function of VPS13B is to directly bridge a cis-Golgi membrane to a trans-Golgi membrane. We believe that either the contacts observed between cis- and trans-Golgi vacuoles generated by hypotonic conditions include some other membranes or that membranes of medial cisterns may (surprisingly, as Golgi membranes are thought to be separate subcompartments) merge in part with cis- membranes and in part with trans-membranes upon exposure to drastic hypotonic conditions. Our new data favor the latter possibility as we find hybrid vacuoles with cis- (Bet1) and medial (Man II) properties and other hybrid vacuoles with medial (Man II) and trans-(GalT) properties. What seems to be clear is that VPS13B selectively accumulates at contacts between two Golgi elements with different properties (and not at contacts between the ER and Golgi complex membranes, in agreement with the lack of an FFAT motif for ER binding via VAP). The spacing between cisterns is compatible with the size of the VPS13B rod as, based on a published cryo EM structure of the Golgi complex (Fig. 2 above), distances between cisterns can be in the range of 25-30nm. In intact cells it is conceivable that VPS13B may reside at both sides of a medial cistern, as suggested by the reviewer and also in agreement with results of the new 4Pi SMS microscopy experiments which show broad localization of VPS13B in the cis-medial Golgi region.

Other points:

Fam177A1 conservation: the 1st paragraph on this protein (p5) would be a good place to describe the full range of organisms that have a homologue: this includes a very wide range of eukaryotes, as shown either from IPR028260. Alignments are informative of the regions that are likely to interact with the conserved VPS13B partner (e.g. in

Capsaspora protein CAOG_09126: aligns best with residues 93-154, including H1 and ~70% of H2, the 1st 12 aa of H2 (KYQYAIDEYYRM) forming a highly conserved amphipathic helix. HHPred additionally finds previously unknown homologues more widely in plants (e.g. At2g18910), which is interesting as A. thaliana has no obvious VPS13B ortholog.

We have now incorporated the following sentence to the manuscript to describe the full range of organisms that have homologue:

“FAM177A1 belongs to a highly conserved FAM177 eukaryotic protein family (Interpro:IPR028260) and based on folding predictions (Jumper et al., 2021; Varadi et al., 2022) FAM177A1 is primarily a disordered protein with two highly conserved α -helices and a short β -sheet hairpin...”

Regarding A. thaliana, as described in Leterme et al., 2023 (PMID: 38033810) and Levine, 2022 (PMID: 36571082) the most common ancestor of Viridiplantae had four VPS13 paralogs: VPS13M, S, B, and Y. VPS13B is lost in several organisms and there are four AtVPS13 paralogs and these are classified as: VPS13X, S, M, and M1. Among the A. thaliana VPS13 paralogs, AtVPS13S (AT5G24740, also known as shrubby) is predicted to be the closest to human VPS13B (Koizumi and Gallagher 2013. PMID: 23444357). Although, there doesn't seem to be a B-sandwich (we refer to this as Jelly Roll Fold in our manuscript) in AT5G24740, the Arabidopsis Information Resource blast predicts hsVPS13B as the strongest homolog for AT5G24740 among the four human paralogs. The evolution of VPS13B is certainly interesting and requires further investigation.

Fam177A1 constructs: I could not find any co-ordinates of these constructs. These are important for others to reproduce the work. Another aspect to consider is how to subdivide the protein rationally. For example, the residues between predicted H1 and H2 126GIST129 are among the most conserved in the whole protein - are they involved with H1's targeting function, or with the function H2 (whatever that is)?

Thank you for pointing out this oversight. We relied on AlphaFold2 prediction to generate different domains.

Helix 1: 118WGPYLWFWYMLRAATSTLSVCDFLGEKIASV147 (or 95W...V124 in isoform 2)

Helix 2: 156QYAIDEYYRMKKEEEEEENRMSEEAQYQQNKLQTDIV197 (133Q...V174 in isoform 2)

Hairpin: 78RRVIHFVSGETME90 (55V...E67 isoform 2)

Our constructs did not include the GIST motif, so we do not think that it is important for membrane localization. We cannot exclude that it may be an important motif for protein-protein interaction, but we can say that it is not involved in helix1 targeting.

Fam177A localization (Fig 2E *now Fig. 2G*): FAM177A1 is membrane bound and it has a wider distribution than both ZPFL-1 and VPS13B. It would be good to know if the FAM177-positive ZPFL1-negative circular profiles are positive for another Golgi marker.

We agree that FAM177A1 distribution is broader in the Golgi complex. FLASH-Paint indicates that FAM177A1 is 21nm away from VPS13B but it does not prefer cis or trans Golgi complex (Fig. 2F).

Minor:

what is the article Legro et al. (in revision)? Can we be told more - at least a title and author list (in a footnote)? Unlike this MS I was unable to find that one in a preprint server.

We had submitted the Legro et al. as an accompanying manuscript during the first submission of the present paper. This manuscript is currently in press in Genetics in Medicine and is now online at (<https://doi.org/10.1016/j.gim.2024.101166>). The first author has now been changed to Kohler, and Legro is now one of the corresponding authors.

Fig 1C-E *now Fig. 1E-G*: the text should explain whether VPS13B is more punctate than the other markers, and also the compare the extent to which label is found outside the Golgi ribbon

The newly added 4Pi-single-molecule switching (SMS) microscopy shows that VPS13B is indeed more punctate than the other Golgi markers (Fig. 1C) in line with what is expected from a membrane contact site protein. This observation is now added to the main text.

Explain how Fig 1F (now Fig. 1H) is not symmetrical X->Y and Y->X

We apologize for the confusion. The heatmap presented in Figure 1F is not symmetric because the median distance analysis is based on a nearest neighbor analysis. It often makes a difference whether we measure a distance from protein A to protein B or from protein B to protein A.

Here is an example: A1 ---- B-----A2

The nearest neighbor for protein A1 is protein B, and the nearest neighbor for protein B is protein A1. In this case, it's a symmetric relationship, and the distance heatmap would be symmetrical. The nearest neighbor for protein A2 is also protein B; however, the nearest neighbor for protein B is still protein A1. This asymmetry presents the reason why the heatmap shown in Figure 1F (now Fig. 1H) is not symmetrical.

Fig 1G (now Fig. 1I) has no scale bar.

Thank you, corrected.

It would help to show the experiments with VPS13B and ER/Golgi markers (line 157).

This comment refers to the hypotonic shock experiment. We have now included images of a representative experiment showing that VPS13B is not at the interface between the ER and GaT (Fig. S1F).

Can Fam177 binding to VPS13B be studied in silico by co-folding all of it (or the likely region - H2) with segments of VPS13B?

We have used Alphafold multimer prediction tool to see if there is any possibility for a FAM177A1 binding. Multimer predicts a potential binding of a β -strand of FAM177A1 to the VAB domain of VPS13B (as shown below, Green indicates VPS13B VAB domain, magenta is full length FAM177A1 iptm_ptm score: 0.6720632679784100 ptm: 0.66968656). However, as this predicted interaction has low probability and we have not found biochemical evidence for a direct interaction of the two proteins, we prefer to omit this prediction. We will include it if we this inclusion was deemed important by the reviewer. We also note that the a.a. sequence of this β -strand is not a region of special conservation within the FAM177A1 sequence.

Fig. 3. Alphafold2 Multimer prediction of VPS13B VAB domain (in green) with FAM177A1. Multimer predicts a new B-sheet in FAM177A1 in the presence of VPS13B.

Fig 2F (now Fig. 2H) bottom row - images wrongly arranged

Thank you, corrected.

Fig. S2G (*now Fig. S2I*): It is not the full picture that "GalT did not colocalize with the remaining VPS13B-positive spots and instead dispersed into the ER as anticipated" (line 239). The few GalT puncta that remain do tend to be at the VPS13B spots.

We modified this sentence to the following: "Moreover, when cells expressing VPS13B and GalT were treated with BFA, GalT dispersed into the ER as anticipated, but some GalT positive spots overlapping with the remaining VPS13B puncta were also observed (Fig. S2G)."

"VPS13B ortholog had not been annotated in zebrafish" (line 277) - though the MS could add that a 3963 aa orthologue has been identified in *Cyprinus carpio* (Common carp) that is over 80% identical to *D rerio* Vps13b.

Thank you for this point. We have now added this information. The text now reads as follows "To test whether our results in cultured cells had a relevance to organism physiology, we carried out studies in zebrafish. Zebrafish has two FAM177 genes, fam177a1a and fam177a1b. In contrast, although Cyprinus carpio encodes a Vps13b, a gene encoding VPS13B ortholog had not been annotated in zebrafish".

Manner of reporting fish results: "indicating a functional interaction" (line 302). The MS could go further: the data show a synthetic phenotype, which aligns with results in Fig 3 and normally indicates a parallel pathway, NOT that the gene products act together as one complex.

Thank you for this comment. We have now replaced the words "indicating a functional interaction" with "indicating a functional partnership".

V minor:

what is the rule by which sometimes Vps13b is written in place of VPS13B?

We refer to the human protein as VPS13B, to the mouse protein as Vps13b and to the zebrafish protein as Vps13b. We used Italics when we are referring to the gene rather than the protein.

June 24, 2024

RE: JCB Manuscript #202311189R

Dr. Pietro V De Camilli
Yale School of Medicine
Cell Biology and Neuroscience
100 College Avenue
room 333
New Haven, CT 06510

Dear Dr. De Camilli:

Thank you for submitting your revised manuscript entitled "VPS13B is localized at the interface between Golgi cisterns and is a functional partner of FAM177A1". We would be happy to publish your paper in JCB pending final revisions necessary to meet our formatting guidelines (see details below).

A. MANUSCRIPT ORGANIZATION AND FORMATTING:

1) Text limits: Character count for Reports is < 20,000, not including spaces. Count includes abstract, introduction, * combined results and discussion, and acknowledgments. Count does not include title page, figure legends, materials and methods, references, tables, or supplemental legends.

2) Figures limits: Reports may have up to 5 main text figures.

3) Figure formatting: * Scale bars must be present on all microscopy images, including inset magnifications. Molecular weight or nucleic acid size markers must be included on all gel electrophoresis. *

4) Statistical analysis: Error bars on graphic representations of numerical data must be clearly described in the figure legend. The number of independent data points (n) represented in a graph must be indicated in the legend. Statistical methods should be explained in full in the materials and methods. For figures presenting pooled data the statistical measure should be defined in the figure legends. Please also be sure to indicate the statistical tests used in each of your experiments (either in the figure legend itself or in a separate methods section) as well as the parameters of the test (for example, if you ran a t-test, please indicate if it was one- or two-sided, etc.). Also, if you used parametric tests, please indicate if the data distribution was tested for normality (and if so, how). If not, you must state something to the effect that "Data distribution was assumed to be normal but this was not formally tested."

5) Abstract and title: The abstract should be no longer than 160 words and should communicate the significance of the paper for a general audience. The title should be less than 100 characters including spaces. Make the title concise but accessible to a general readership.

6) Materials and methods: Should be comprehensive and not simply reference a previous publication for details on how an experiment was performed. Please provide full descriptions in the text for readers who may not have access to referenced manuscripts.

7) All antibodies, cell lines, animals, and tools used in the manuscript should be described in full, including accession numbers for materials available in a public repository such as the Resource Identification Portal. Please be sure to provide the sequences for all of your primers/oligos and RNAi constructs in the materials and methods. You must also indicate in the methods the source, species, and catalog numbers (where appropriate) for all of your antibodies. Please also indicate the acquisition and quantification methods for immunoblotting/western blots.

8) Microscope image acquisition: The following information must be provided about the acquisition and processing of images:

- a. Make and model of microscope
- b. Type, magnification, and numerical aperture of the objective lenses
- c. Temperature
- d. Imaging medium

- e. Fluorochromes
- f. Camera make and model
- g. Acquisition software
- h. Any software used for image processing subsequent to data acquisition. Please include details and types of operations involved (e.g., type of deconvolution, 3D reconstitutions, surface or volume rendering, gamma adjustments, etc.).

10) Supplemental materials: There are strict limits on the allowable amount of supplemental data. Articles may have up to 5 supplemental figures. Please also note that tables, like figures, should be provided as individual, editable files. A summary of all supplemental material should appear at the end of the Materials and methods section.

13) ORCID IDs: ORCID IDs are unique identifiers allowing researchers to create a record of their various scholarly contributions in a single place. Please note that ORCID IDs are now *required* for all authors. At resubmission of your final files, please be sure to provide your ORCID ID and those of all co-authors.

Please note that JCB now requires authors to submit Source Data used to generate figures containing gels and Western blots with all revised manuscripts. This Source Data consists of fully uncropped and unprocessed images for each gel/blot displayed in the main and supplemental figures. Since your paper includes cropped gel and/or blot images, please be sure to provide one Source Data file for each figure that contains gels and/or blots along with your revised manuscript files. File names for Source Data figures should be alphanumeric without any spaces or special characters (i.e., SourceDataF#, where F# refers to the associated main figure number or SourceDataFS# for those associated with Supplementary figures). The lanes of the gels/blots should be labeled as they are in the associated figure, the place where cropping was applied should be marked (with a box), and molecular weight/size standards should be labeled wherever possible.

Journal of Cell Biology now requires a data availability statement for all research article submissions. These statements will be published in the article directly above the Acknowledgments. The statement should address all data underlying the research presented in the manuscript. Please visit the JCB instructions for authors for guidelines and examples of statements at (<https://rupress.org/jcb/pages/editorial-policies#data-availability-statement>).

B. FINAL FILES:

****It is JCB policy that if requested, original data images must be made available to the editors. Failure to provide original images upon request will result in unavoidable delays in publication. Please ensure that you have access to all original data images prior to final submission.****

****The license to publish form must be signed before your manuscript can be sent to production. A link to the electronic license to publish form will be sent to the corresponding author only. Please take a moment to check your funder requirements before choosing the appropriate license.****

Thank you for your attention to these final processing requirements. Please revise and format the manuscript and upload materials within 7 days. If you need an extension for whatever reason, please let us know and we can work with you to determine a suitable revision period.

Thank you for this interesting contribution, we look forward to publishing your paper in Journal of Cell Biology.

Sincerely,

Jodi Nunnari, Ph.D.
Editor-in-Chief

Andrea L. Marat, Ph.D.
Senior Scientific Editor

Journal of Cell Biology

Reviewer #1 (Comments to the Authors (Required)):

The Authors have satisfactorily addressed my comments.

Reviewer #2 (Comments to the Authors (Required)):

This manuscript presents novel findings on the localization and role of VPS13B at the Golgi apparatus. Interestingly, the authors also report on a VPS13B interacting protein, FAM177A1, which appears to share some functions with VPS13B. Super-resolution microscopy (FLASH-PAINT) data has been exploited to report on the relative distances between VPS13B, FAM177A1, and a number of Golgi markers. This, together with hypotonic shock treatment-based experiments, indicate that VPS13B (and FAM177A1) localizes between Golgi cisternae. They later found that these two proteins are required for a fast reformation of the Golgi after BFA-induced disassembly.

In this revised version, all my comments and questions have been clearly addressed, either by providing new experimental data or by clarifying it in the review response and/or in the text. The current version of the manuscript stands on its own, and the conclusions drawn are strongly supported by the data, so I am very happy to endorse this manuscript for publication at JCB.

Maybe only a very minor thing, the new title talks about "Golgi cisterns", which is relatively uncommon wording in the field, I'd rather suggest the authors to use the, I think, more common term "Golgi cisternae".

Reviewer #3 (Comments to the Authors (Required)):

I am happy with the revisions. I agree with the authors that the AlphaFold model of VPS13B and FAM177A1 is unreliable. I carried out a similar piece of modelling in AF3 and there was no consistency, which if present might have been worth reporting.